# *Cis*-regulatory modes of *Ultrabithorax* inactivation in butterfly forewings

Amruta Tendolkar[1], Anyi Mazo-Vargas[1], Luca Livraghi[1], Joseph J Hanly[1,2], Kelsey C Van Horne[1], Lawrence E Gilbert[3], Arnaud Martin[1]*

[1]Department of Biological Sciences, The George Washington University, Washington, DC, United States; [2]Smithsonian Tropical Research Institute, Panama City, Panama; [3]Department of Integrative Biology, University of Texas – Austin, Austin, United States

*For correspondence:
arnaud@gwu.edu

Competing interest: The authors declare that no competing interests exist.

**Abstract** *Hox* gene clusters encode transcription factors that drive regional specialization during animal development: for example the Hox factor Ubx is expressed in the insect metathoracic (T3) wing appendages and differentiates them from T2 mesothoracic identities. *Hox* transcriptional regulation requires silencing activities that prevent spurious activation and regulatory crosstalks in the wrong tissues, but this has seldom been studied in insects other than *Drosophila*, which shows a derived *Hox* dislocation into two genomic clusters that disjoined *Antennapedia* (*Antp*) and *Ultrabithorax* (*Ubx*). Here, we investigated how *Ubx* is restricted to the hindwing in butterflies, amidst a contiguous *Hox* cluster. By analysing Hi-C and ATAC-seq data in the butterfly *Junonia coenia*, we show that a Topologically Associated Domain (TAD) maintains a hindwing-enriched profile of chromatin opening around *Ubx*. This TAD is bordered by a Boundary Element (BE) that separates it from a region of joined wing activity around the *Antp* locus. CRISPR mutational perturbation of this BE releases ectopic *Ubx* expression in forewings, inducing homeotic clones with hindwing identities. Further mutational interrogation of two non-coding RNA encoding regions and one putative *cis*-regulatory module within the *Ubx* TAD cause rare homeotic transformations in both directions, indicating the presence of both activating and repressing chromatin features. We also describe a series of spontaneous forewing homeotic phenotypes obtained in *Heliconius* butterflies, and discuss their possible mutational basis. By leveraging the extensive wing specialization found in butterflies, our initial exploration of *Ubx* regulation demonstrates the existence of silencing and insulating sequences that prevent its spurious expression in forewings.

## eLife assessment

This **valuable** paper examines the Bithorax complex in several butterfly species, in which the complex is contiguous and not split, as it is in the well-studied fruit fly *Drosophila*. Based on genetic screens and genetic manipulations of a boundary element involved in segment-specific regulation of Ubx, the authors provide **convincing** evidence for their conclusions, which could be strengthened by additional data and analyses in the future. The data presented are relevant for those interested in the evolution and function of Hox genes and of gene regulation in general.

## Introduction

*Hox* genes are key specifiers of antero-posterior regional identity in animals, and thus require robust regulatory mechanisms that confine their expression to well-delimited sections of the body (*Lewis, 1978*). Their genomic arrangement into *Hox* gene clusters has provided a rich template for the study of gene regulation, with mechanisms including chromatin silencing and opening, 3D conformational

changes, and non-coding RNAs (*Mallo and Alonso, 2013*). However, this rich body of work has been almost exclusively performed in mice and fruit flies. In order to decipher how diverse body plans and morphologies evolved, we must start assessing the mechanisms of Hox gene regulation in a wider range of organisms.

The *Ultrabithorax* (*Ubx*) gene encodes a Hox family transcription factor involved in the specification of segment identities in arthropods (*Hughes and Kaufman, 2002*; *Heffer and Pick, 2013*). In insects, the conserved expression of *Ubx* in the metathoracic (T3) segment is required for their differentiation from Ubx-free tissues in the mesothorax (T2), and has been a key factor for the specialization of metathoracic serial appendages including T3 legs (*Mahfooz et al., 2007*; *Refki et al., 2014*; *Tomoyasu, 2017*; *Feng et al., 2022*; *Buffry et al., 2023*) and hindwings or their derivatives (*Tomoyasu, 2017*; *Loker et al., 2021*). The mechanisms of *Ubx* segment-specific expression have been intensively studied in *D. melanogaster* (*Mallo and Alonso, 2013*; *Hajirnis and Mishra, 2021*), where *Hox* genes are separated into two genomic loci, the Antennapedia (ANT-C, *Antp*) and Bithorax clusters (BX-C). In short, the BX-C complex that includes *Ubx*, *abdominal-A* (*abd-A*), and *Abdominal-B* (*Abd-B*) is compartmentalized into nine chromosomal domains that determine the parasegmental expression boundaries of these three genes (*Maeda and Karch, 2015*). For instance, the deletion of a small region situated between *Ubx* and *abd-A* produces the *Front-ultraabdominal* phenotype (*Fub*) where the first abdominal segment (A1) is transformed into a copy of the second abdominal segment A2, due to a gain-of-expression of *abd-A* in A1 where it is normally repressed (*Pease et al., 2013*). At the molecular level, the *Fub* boundary is enforced by insulating factors that separate Topologically Associating Domains (TADs) of open-chromatin, while also allowing interactions of *Ubx* and *abd-A* enhancers with their target promoters (*Postika et al., 2018*; *Srinivasan and Mishra, 2020*). Likewise, the *Fab-7* deletion, which removes a TAD boundary insulating *abd-A* and *Abd–B* (*Moniot-Perron et al., 2023*), transforms parasegment 11 into parasegment 12 due to an anterior gain-of-expression of *Abd-B* (*Gyurkovics et al., 1990*). By extrapolation, one may expect that if the *Drosophila Hox* locus was not dislocated into two complexes, Antp and Ubx 3D contact domains would be separated by a Boundary Element (BE), and that deletions similar with *Fub* and *Fab-7* mutations would result in gain-of-function mutations of *Ubx* that could effectively transform T2 regions into T3 identities. The BX-C locus also includes non-coding RNAs (*Pease et al., 2013*), some of which are processed into miRNAs known to repress *Ubx* and *abd-A* (*Gummalla et al., 2012*; *Garaulet and Lai, 2015*). *Fub-1/bxd* long non-coding RNAs (lncRNAs) situated 5' of *Ubx* are thought to participate in *Ubx* regulation in the PS5 (posterior T3 to anterior A1) parasegment (*Petruk et al., 2006*; *Ibragimov et al., 2023*). An intronic lncRNA dubbed *lncRNA:PS4* is expressed in the PS4 parasegment (posterior T2 - anterior T3), and appears to stabilize *Ubx* in this region in mutant contexts (*Hermann et al., 2022*). Little is known about how insect *Hox* genes are regulated outside of *Drosophila*, where they co-localize into a single *Hox* cluster, and where *Antp* and *Ubx* thus occur in contiguous positions (*Gaunt, 2022*; *Mulhair and Holland, 2024*). A few *Hox*-related miRNAs are evolutionarily conserved across the locus in arthropods (*Pace et al., 2016*), and an early study in *Tribolium* characterized the embryonic expression of a *Hox* cluster non-coding transcripts (*Shippy et al., 2008*).

These knowledge gaps lead us to consider the use of butterflies and moths (Lepidoptera) as alternative model systems for the study of *Ubx* function and regulation. Lepidopteran forewings and hindwings are functionally and morphologically differentiated, and CRISPR mosaic knock-outs (mKOs) showed that *Ubx* is necessary for the specification of hindwing color patterns, shape, and venation (*Tendolkar et al., 2021*). In three species of nymphalid butterflies (*Heliconius erato*, *Junonia coenia*, and *Bicyclus anynana*), CRISPR-mediated loss-of-function of *Ubx* induces regional-specific homeotic transformations of hindwing patterns into their forewing counterpart (*Matsuoka and Monteiro, 2018*; *Tendolkar et al., 2021*), reminiscent of homeotic aberrations that are sporadically observed in butterfly wings (*Sibatani, 1983*; *Nijhout and Rountree, 1995*). The ectopic activation of *Ubx* into the pupal forewing results in the gain of hindwing features, suggesting Ubx is sufficient to drive T3-like identity when expressed in T2 (*Lewis et al., 1999*; *Tong et al., 2014*). Besides its roles in adult wing differentiation, *Ubx* is also known to repress thoracic leg identity in transient embryonic appendages of the first abdominal segment, called pleuropods (*Zheng et al., 1999*; *Masumoto et al., 2009*; *Tong et al., 2017*; *Tendolkar et al., 2021*; *Matsuoka et al., 2022*). The general organization of Hox gene clusters has been well described in Lepidoptera, but their regulation has been seldom studied. Lepidopteran genomes have accumulated divergent *Hox3* copies, named *Shox* genes, that are required

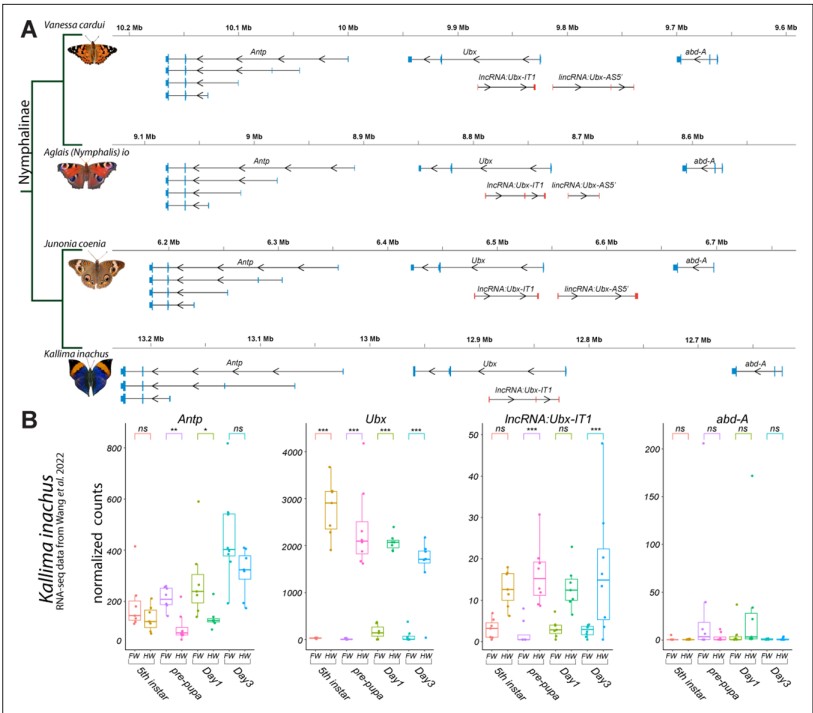

**Figure 1.** Annotation of the *Ubx* genomic interval in four butterflies of the Nymphalinae sub-family. (**A**) Genomic intervals spanning *Antp*, *Ubx*, and *abd-A*, featuring published transcript annotations from NCBI Reference Genomes of *V. cardui* and *A. io*, and manual re-annotations of the *J. coenia* and *K. inachus* genomes using published RNAseq dataset (see Methods). Exons are shown with coding (thick) and non-coding (thin) sections. No *lincRNA:Ubx-AS5'* transcripts were detected in *K. inachus*. (**B**) Expression profiling of transcripts of the *Ubx* region in *K. inachus*, based on a reanalysis of published wing RNA-seq transcriptomes (***Wang et al., 2022***). Expression levels are plotted as DESeq2 normalized counts plots. Pairwise Wald tests adjusted for multiple test correction each assess differential expression between forewings and hindwings. ns: non-significant; *: p<0.05; **: p<0.01; ***: p<0.001.

during early embryonic development but do not appear to play homeotic functions (***Ferguson et al., 2014***; ***Livraghi, 2017***; ***Mulhair et al., 2023***). A lncRNA and two miRNAs were identified in the intergenic region between *abd-A* and *Abd-B* in the silkworm (***Wang et al., 2017***; ***Wang et al., 2019***). In butterfly wings, the regulation of *Ubx* shows strong patterns of segment-specific regulation at two levels. First, the *Ubx* transcript has been consistently identified as the most differentially expressed mRNA between the two wing sets (***Hanly et al., 2019***; ***Wang et al., 2022***). Second, comparison of ATAC-seq signals reveal that forewing vs. hindwing have identical open-chromatin profiles during development across the genome, except at the *Ubx* gene itself (***Lewis et al., 2019***; ***van der Burg et al., 2019***). Thus, the ability of the *Ubx* locus to be robustly activated in hindwings and repressed in forewings is likely driving most subsequent differences between these tissues. In this study, we provide an initial assessment of the regulation of the *Ubx* locus during butterfly wing development. To do this, we leverage genomic resources and CRISPR mutagenesis with a focus on two laboratory species belonging to the Nymphalinae sub-family, *J. coenia* and *Vanessa cardui* (***Livraghi et al., 2017***; ***Martin et al., 2020***; ***van der Burg et al., 2020***; ***Mazo-Vargas et al., 2022***). We identify putative regulatory regions likely involved in the repression and activation of *Ubx* expression, and discuss the potential mechanisms restricting it to hindwings. Finally, we describe a collection of spontaneous wing homeotic mutants in *Heliconius spp.* and elaborate on the categories of mutations that could underlie these phenotypes by misregulating *Ubx*.

## Results

### Gene expression at the *Ubx* locus during wing development

We provide annotations of the *Ubx* genomic region in four Nymphalinae butterflies (*Figure 1A*). These feature existing genomic resources for our model species *J. coenia* and *V. cardui* (*van der Burg et al., 2020*; *Lohse et al., 2021b*; *Zhang et al., 2021*), as well as for *Aglais* (*Nymphalis*) *io* (*Lohse et al., 2021a*). The publicly available annotations for these three species include evidence from developmental transcriptomes, and we added to this set a manual annotation of the *Ubx* locus from the oak leaf butterfly *Kallima inachus*, for which forewing *vs.* hindwing transcriptomes have been sequenced across a replicated developmental time series (*Yang et al., 2020*; *Wang et al., 2022*).

All Nymphalinae show a similar organization of the region spanning *Ubx*. Interestingly, the first intron of *Ubx* encodes a long non-coding RNA in opposite orientation to *Ubx*, that we dub here *lncRNA:Ubx-IT1* (abbr. *Ubx-IT1*), based on the recommended nomenclature (*Seal et al., 2023*). Orthologous versions of *Ubx-IT1* are detected in most NCBI RefSeq genome annotations throughout Lepidoptera (*e.g.* the ncRNA *NCBI:XR_960726* in *Plutella xylostella*), implying it is a conserved feature of the *Ubx* locus in this insect order. Finally, both annotations from *V. cardui, A. io,* and *J. coenia* show a long intergenic non-coding transcript starting in antisense orientation about 10–15 kb 5' of *Ubx*, that we dub here *lincRNA:Ubx-AS5'* (abbr. *Ubx-AS5'*). This transcript was neither detected in *K. inachus* transcriptomes nor in RNA datasets outside of the Nymphalinae sub-family, and could be specific to this lineage. Next we reanalyzed the *K. inachus* wing transcriptomes (*Wang et al., 2022*), and profiled the expression of *Ubx* region transcripts during wing development (*Figure 1B*). As expected from previous studies (*Hanly et al., 2019*; *Paul et al., 2021*; *Merabet and Carnesecchi, 2022*; *Wang et al., 2022*), *Ubx* showed a strong expression bias in hindwings, spanning the larval imaginal disks to the intermediate pupal stage. Of note, Ubx is confined to the peripodial membranes of insect T2 wing appendages (*Weatherbee et al., 1998*; *Weatherbee et al., 1999*; *Prasad et al., 2016*), which may explain residual detection in some of the forewing samples here. *Ubx-IT1* was significantly enriched in hindwings compared to forewings, albeit at ~100-fold lower count levels than *Ubx* in the same samples. The *Hox* gene *Antp* showed a minor forewing enrichment, confirming that while *Ubx* expression is robustly repressed in T2 forewing tissues, *Antp* expression is permitted in both T2 and T3 appendages (*Matsuoka and Monteiro, 2021*; *Matsuoka and Monteiro, 2022*; *Paul et al., 2021*). Expression of *abd-A* was undetected in most wing samples.

### Chromatin 3D conformation reveals a Boundary Element between *Antp* and *Ubx*

Genome-wide Hi-C sequencing can be used to generate heatmaps that capture the conformation of 3D chromatin in tissues, and has been used extensively to study *Drosophila Hox* cluster organisation into TADs that prevent regulatory crosstalk between adjacent genes (*Ibragimov et al., 2023*; *Moniot-Perron et al., 2023*). Here, we used Hi-C to assess the 3D chromatin architecture of the *Hox* cluster interval in the butterfly *J. coenia*, using existing datasets that were generated from fifth instar larval forewings (*van der Burg et al., 2020*; *Mazo-Vargas et al., 2022*). In larval forewings, the *Hox* chromatin conformation landscape consists of three well-delimited TADs, the first one spanning *proboscipedia* (*pb*) to *Sex comb reduced* (*Scr*), the second one around *Antp*, and the third one *Ubx, abd-A,* and *Abd-B* (*Figures 2 and 3A*). A Boundary Element (BE), was robustly called (see Materials and methods) at the region separating the *Antp* and *Ubx* TADs, situated in the *Ubx* last intron. Because TAD boundary prediction has a coarse resolution, we arbitrarily define the candidate BE region as a 15 kb interval centered in the *Ubx* last intron, and dub it *Antp-Ubx_BE*. A binding motif analysis identified 4 CTCF binding sites in a 1 kb interval within *Antp-Ubx_BE*, two of which were found in a tightly linked, convergent orientation (*Figure 2—figure supplement 1*), which is consistent with TAD insulating role in mediating chromatin loop-extrusion (*Guo et al., 2015*). This concordance between Hi-C profiling and CTCF motif prediction thus suggests that *Antp-Ubx_BE* region might function as an insulator between regulatory domains of *Antp* and *Ubx*.

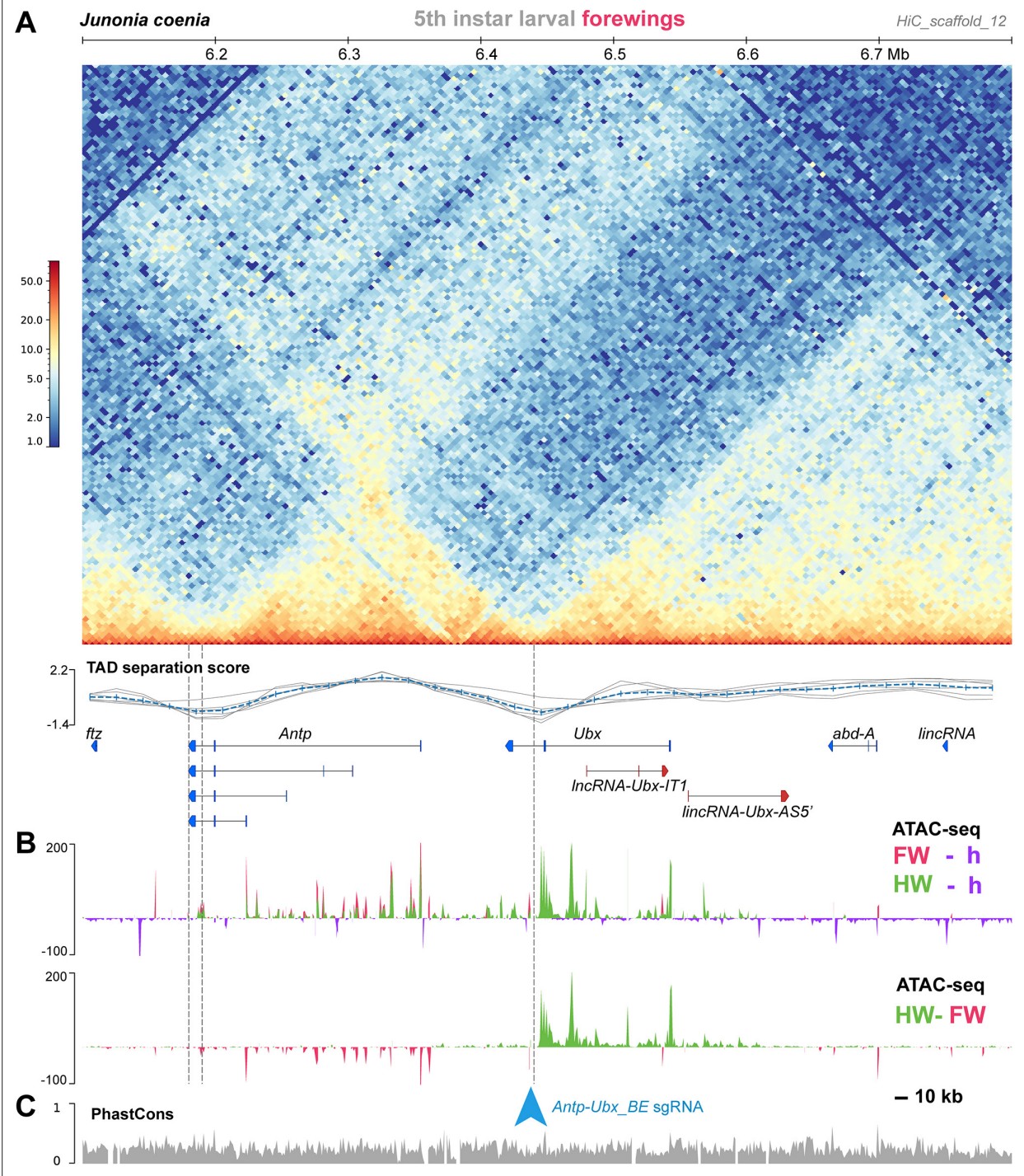

**Figure 2.** A region of hindwing-specific chromatin-opening is bordered by a TAD BE in the last intron of *Ubx*. (**A**) Hi-C contact heatmap in fifth instar forewings of *J. coenia* and TAD separation scores around *Ubx*. A TAD boundary element (*Antp-Ubx_BE*) is inferred in the last intron of *Ubx* (vertical dotted line). (**B**) Differential ATAC-seq profiles, re-analyzed from a previous dataset (***Mazo-Vargas et al., 2022***). Top: open-chromatin profiles of forewings (FW, magenta), and hindwings (HW, green), respectively subtracted from larval head signal (purple, negative when wing signals at background-level). Bottom: subtractive ATAC-seq profile (HW-FW) revealing hindwing-enriched chromatin in the *Ubx* locus. *Antp-Ubx_BE* is in the vicinity of an isolated region of forewing-enriched opening (blue arrowhead). (**C**) *PhastCons* genomic alignment scores, with overall alignability suggesting minimal structural variation across this interval in Lepidoptera and Trichoptera.

The online version of this article includes the following figure supplement(s) for figure 2:

**Figure supplement 1.** Prediction of two conserved CTCF binding sites at *Antp-Ubx_BE*.

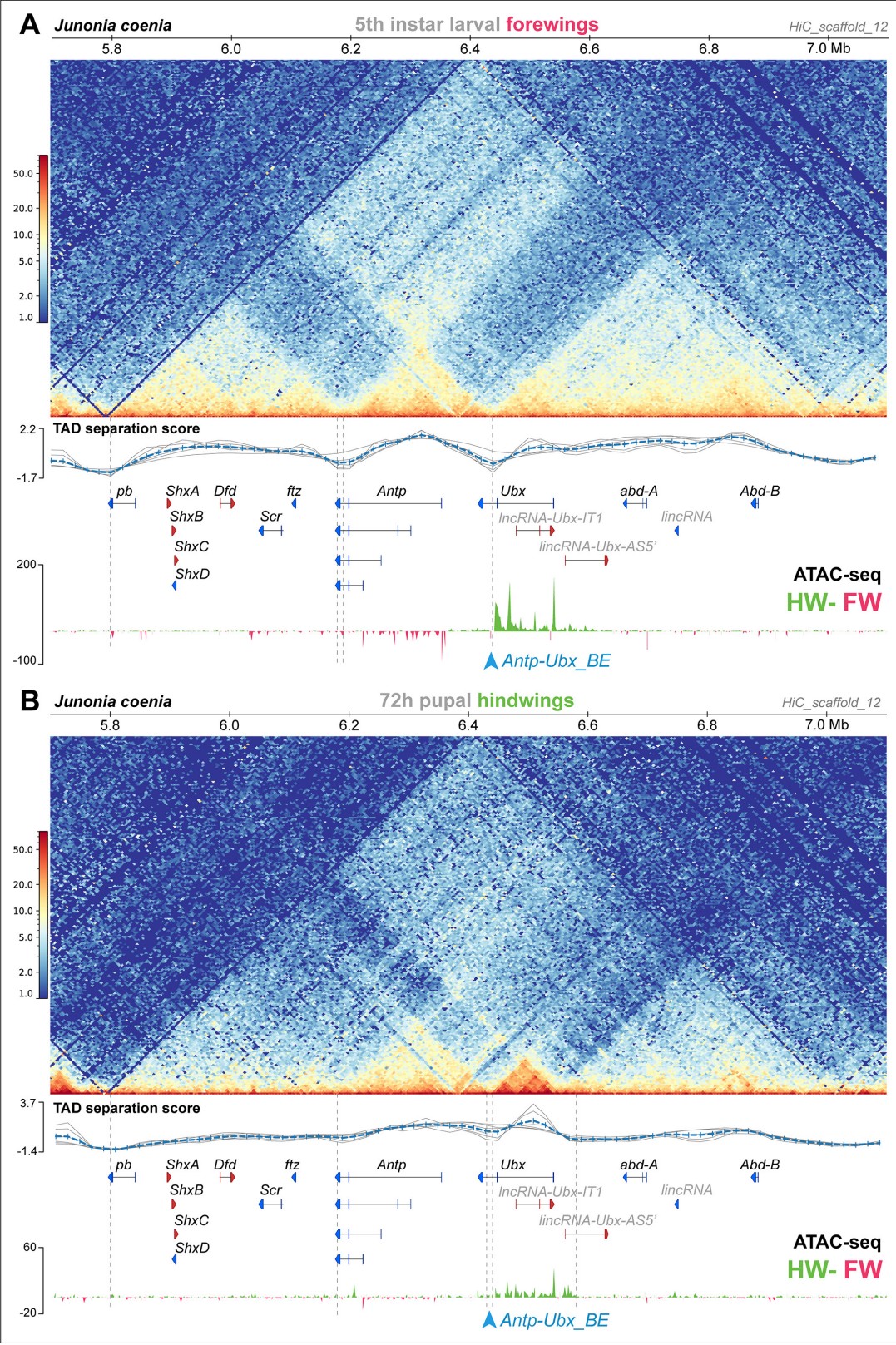

**Figure 3.** Hindwing-enriched chromatin-opening around *Ubx*, and the *Antp-Ubx_BE* boundary, are both maintained in mid-pupal hindwings. (**A**) Hi-C heatmap in *J. coenia* fifth instar larval forewings, and subtractive ATAC-seq profiles at this stage (hindwing-forewing), as expanded from *Figure 2* across the *Hox* cluster. (**B**) Hi-C heatmap in *J. coenia* mid-pupal hindwings, and subtractive ATAC-seq profiles at this stage (forewing-hindwing). Inferred TAD boundaries are shown as vertical dotted lines. Blue arrowhead: position of the *Antp-Ubx_BE* sgRNA.

## Differential forewing vs. hindwing chromatin-opening across the *Antp-Ubx* interval

In flies, the *Ubx/abd-A* section is organized into regulatory domains that display differential activities across the antero-posterior axis, following what has been called the open-for-business model (*Maeda and Karch, 2015*; *Gaunt, 2022*). Here we tested if this pattern extends to butterfly species with a contiguous *Hox* cluster. To do this, we used ATAC-seq datasets from *J. coenia* forewing (T2), hindwing (T3), and whole-head tissues sampled across fifth instar larval and early pupal stages, similarly to previous studies (*van der Burg et al., 2020*; *Mazo-Vargas et al., 2022*; *Van Belleghem et al., 2023*). These data reveal that chromatin opening along the *Antp/Ubx/abd-A* interval is partitioned into several regions showing a transition of T2 to T3 activity (*Figure 2B*). From the anterior to posterior *Hox* collinear order (i.e. from *Antp* towards *abd-A*), chromatin-opening forms a block of forewing-enriched activity close to *Antp* and its 5' region, to a block of activity in both forewings and hindwings that stops at the *Antp-Ubx_BE*. This region is consistent with the fact that *Antp* is expressed in both wing pairs (*Figure 1B*). From *Antp-Ubx_BE*, the interval including *Ubx* and a large upstream region is strongly enriched for hindwing opening, consistently with previous studies that found it to be the only genomic region showing this pattern (*Lewis et al., 2019*; *van der Burg et al., 2019*). Last, the region surrounding *abd-A* is devoid of differential open-chromatin activity between forewings and hindwings, in accordance with the exclusion of its expression from thoracic segments (*Warren et al., 1994*; *Tong et al., 2014*).

## Comparison of 3D conformation and open-chromatin profiles between larval forewings and mid-pupal hindwings

The Hi-C dataset analyzed above was prepared from larval forewings, and forewings do not express Ubx (*Figure 1B*). Next, we repeated our analysis on a Hi-C dataset generated in pupal hindwings instead (*van der Burg et al., 2020*), that is in a later-stage tissue expressing *Ubx*. We found two main differences in this contact landscape compared to the larval forewing (*Figure 3*). First, the TAD spanning from *proboscipedia* (*pb*) to *fushi-tarazu* (*ftz*) faded in intensity, while in contrast, the TAD around *Antp* remained strongly organized. Second, *Ubx* lost its physical association to the *abd-A* and *Abd-B* domains, and gained a TAD boundary situated in the *Ubx-AS5'* intron. It is difficult to disentangle effects from staging (larval vs. pupal) and tissues (forewing vs. hindwing) in this comparison. Specifically, these differences we observed may be due to chromatin remodeling between stages, as somewhat expected during metamorphosis (*Gutierrez-Perez et al., 2019*). Alternatively, it is also possible hindwing development requires *Ubx* to be insulated from the more posterior enhancers. These issues will require further investigation, for instance using profiling of histone marks, with pairwise forewing-hindwing comparison at single stages. Nonetheless the later hindwing sample showed a maintenance of *Antp-Ubx* separation. First, while *Ubx* formed a smaller TAD spanning its coding exons 1–2, this region conserved a domain of hindwing-enriched open-chromatin. Second, boundary prediction called two possible, tightly linked TAD limits in the *Antp-Ubx_BE* region, showing that the last intron of *Ubx* still acts as an insulating region. In conclusion, our preliminary comparison of *Hox* 3D conformation indicates that the *Antp-Ubx_BE* is relatively stable across two stages and wing serial homologs.

## Mutagenic perturbation of *Antp-Ubx_BE* results in forewing homeosis

Next, we reasoned that the forewing-enriched ATAC-seq peak present in the inferred boundary interval (*Figure 4A*) might mediate the binding of insulator proteins (*Savitsky et al., 2016*; *Stadler et al., 2017*), or act as a transcriptional silencer (*Segert et al., 2021*). Several genomic features support the former hypothesis. First, the only forewing-enriched ATAC-seq peak across a 150 kb region (spanning the *Ubx* gene and the *Antp-Ubx* intergenic region), coincides with the midpoint between the two tentative *hicFindTADs* boundary predictions inferred from HiC data (*Figure 2B*). Second, during motif scans conducted across that 150 kb region we found eight predicted binding-sites for the *Drosophila* CCCTC-Binding Factor (CTCF) clustered in a 5 kb region around the differentially accessible region, and none elsewhere in the last Ubx intron (*Figure 4A*), suggesting the forewing-enriched ATAC-seq peak may function as a transcriptional insulator (*Gambetta and Furlong, 2018*; *Postika et al., 2018*; *Kyrchanova et al., 2020*; *Kaushal et al., 2022*). Last, the two candidate CTCF binding motifs that are within the forewing-enriched ATAC-seq peak are also conserved across Lepidoptera and Trichoptera

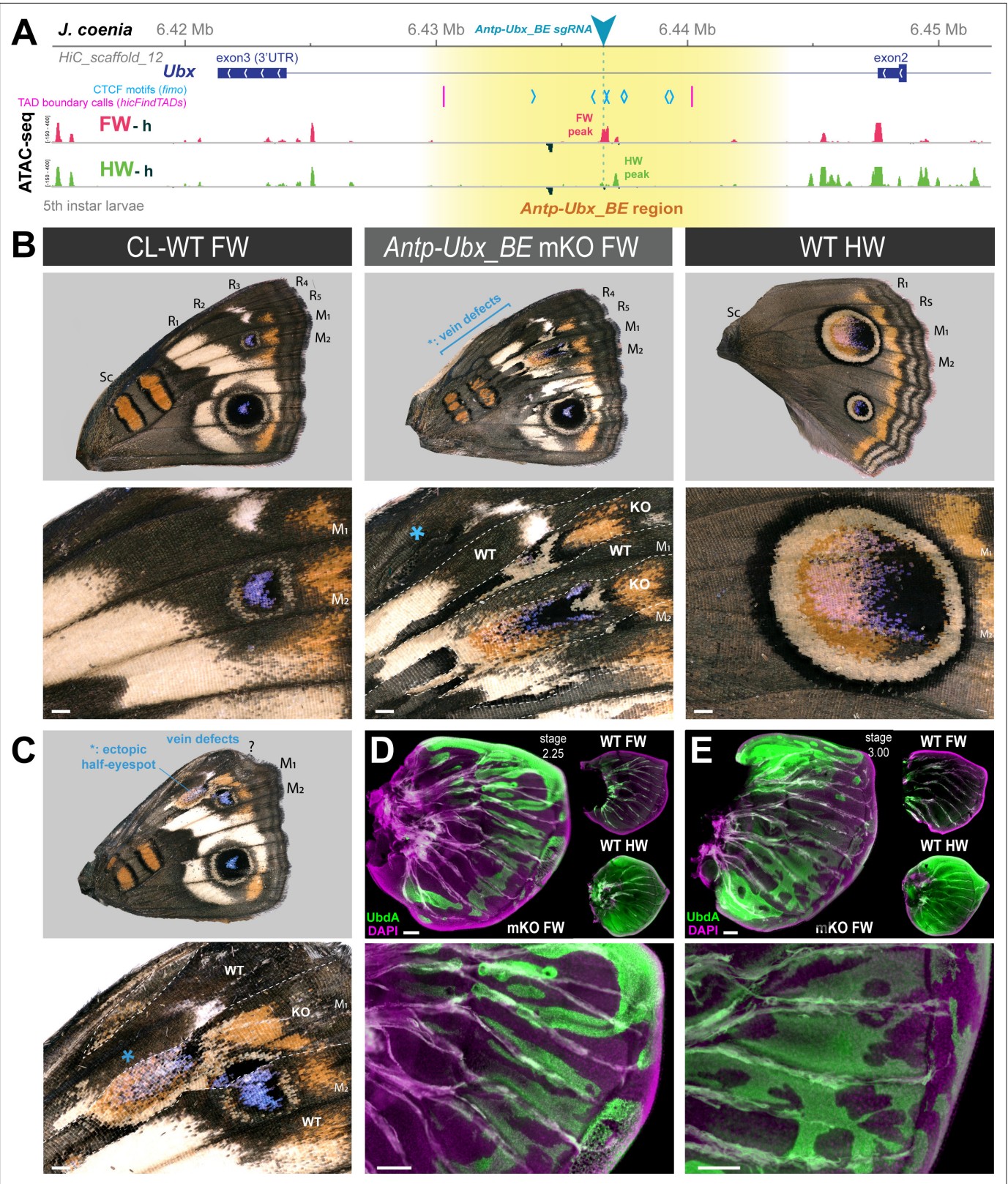

**Figure 4.** CRISPR perturbation of *Antp-Ubx_BE* results in FW➔HW homeoses. (**A**) *Antp-Ubx_BE* sgRNA targeting (cyan triangle) of a FW-enriched ATAC-peak (magenta) within the *Ubx* last intron. (**B–C**) Two examples of *J. coenia Antp-Ubx_BE* crispants showing mosaic FW➔HW homeoses, shown in dorsal views. CL-WT: contralateral, horizontally flipped images of forewings from the same individuals. WT HW: wild type hindwings from the same individual and mutant forewing side. Both individuals show disruption of their Radial veins (R₁-R₅ area). The specimen shown in C displays a partial,

*Figure 4 continued on next page*

*Figure 4 continued*

ectopic eyespot (asterisk). (**D–E**) Immunofluorescent detection of the UbdA epitope (green) in fifth instar wings disks of *Antp-Ubx_BE* crispants, revealing ectopic antigenicity in forewings. WT forewings of similar stage, and HW from the same crispant individuals, are shown for comparison as insets. Green autofluorescence was observed in tracheal tissues. Scale bars: B-C = 500 μm; D-E = 100 μm.

The online version of this article includes the following figure supplement(s) for figure 4:

**Figure supplement 1.** CRISPR perturbation of the *Antp-Ubx* boundary element results in FW-to-HW homeosis.

**Figure supplement 2.** Pupal defects following FW→HW homeosis in *Antp-Ubx_BE* crispants.

**Figure supplement 3.** Validation of CRISPR-induced DNA lesions in an *Antp-Ubx_BE* crispant pupal forewing.

**Figure supplement 4.** Additional examples of ectopic UbdA and FW→HW homeosis in *Antp-Ubx_BE* crispant larval forewings.

(*Figure 2—figure supplement 1*), two lineages that diverged around 300 Mya (*Jusino et al., 2019*; *Thomas et al., 2020*).

To test this hypothesis, we used CRISPR targeted mutagenesis to perturb *Antp-Ubx_BE* and assess its functionality, and designed a single sgRNA in a conserved sequence within the forewing-enriched ATAC-seq (*Figure 2—figure supplement 1*). Remarkably, CRISPR mutagenesis of the *Antp-Ubx_ BE* target induced $G_0$ mutants with homeotic transformations of their forewings into hindwings (*Figure 4B–C* and *Figure 4—figure supplement 1*), including identity shifts in patterns, venation, and wing shape. It is important to note that none of the resulting crispants showed hindwing effects. Thus, we can reasonably attribute forewing homeotic phenotypes to indel mutations restricted to the intronic region, without disruption of the *Ubx* transcript, as this would result in hindwing phenotypes (*Matsuoka and Monteiro, 2021*; *Tendolkar et al., 2021*).

Homeotic clones are first visible in *Antp-Ubx_BE* crispants at the pupal stage, with streaks of thinner cuticle, sometimes associated with local necrosis or with suture defects in the ventral midline, in particular where leg and wing pouches adjoin (*Figure 4—figure supplement 2*). When dissected out of the pupa, these mutant forewings also show streaks of more transparent wing epithelium, concomitantly with the territories of thinner cuticle above, and PCR genotyping of affected pupal wing tissue confirmed the presence of CRISPR-induced mutations at the *Antp-Ubx_BE* target site (*Figure 4—figure supplement 3*). Color pattern homeotic clones were salient at the adult stage, with for example, clonal losses of the forewing specific white-band, and partial acquisitions of the large $M_1$-$M_2$ hindwing eyespot. In one specimen, an ectopic, partial $M_1$-$M_2$ hindwing eyespot appeared in the $R_5$-$M_1$ region, suggesting a perturbation of the eyespot induction process in this wing. Nymphalid forewings have five radial veins ($R_{1-5}$), which provide sturdiness for flight (*Wootton, 1993*), while hindwings have only two Radial veins. Forewing homeotic mutants showed mosaic venation defects in the Radial vein area (*Figure 4B*). Finally, higher expressivity mutant forewings were smaller and rounder, reminiscent of hindwing shape.

Next, we dissected fifth instar larval wing disks from developing *Antp-Ubx_BE* crispants, and monitored the expression of Ubd-A (Ubx and Abd-A epitopes), normally restricted to the hindwing and only present in the forewing peripodial membrane (*Weatherbee et al., 1999*). Crispants showed forewing clones with strong ectopic expression of Ubd-A (*Figure 4D–E* and *Figure 4—figure supplement 4*). This result supports the inference that *Antp-Ubx_BE* forewing homeoses are due to the de-repression of *Ubx* in this tissue.

## Mutational interrogation of lncRNA-encoding regions at the *Ubx* locus

We used CRISPR mutagenesis to test the function of the two lncRNA-encoding loci at the *Ubx* locus. Mutagenesis of the *Ubx-IT1* first exon in *J. coenia*, and of the *Ubx-IT1* promoter in *V. cardui*, both resulted in crispants with small homeotic phenotypes in forewings and hindwings (*Figure 5* and *Figure 5—figure supplement 1*). This result contrasts with *Ubx* exon mKO experiments, which only generate hindwing phenotypes (*Tendolkar et al., 2021*). Given the scarcity of *Ubx-IT1* crispants obtained (11 out of 236 adults), and the small size of the homeotic clones within them, we infer the occasional phenotypes may be due to rare alleles. Thus, rather than evidence of functionality of the *Ubx-IT1* transcript, the homeotic phenotypes may rather reflect the effects of regulatory perturbation on *Ubx* itself, with some random mutations in this intronic region resulting in hindwing *Ubx* loss-of-function, and some others triggering derepression in forewings. Likewise, next we mutagenized the first exon of *Ubx-AS5'*, located upstream of the *Ubx* promoter, and obtained twelve hindwing

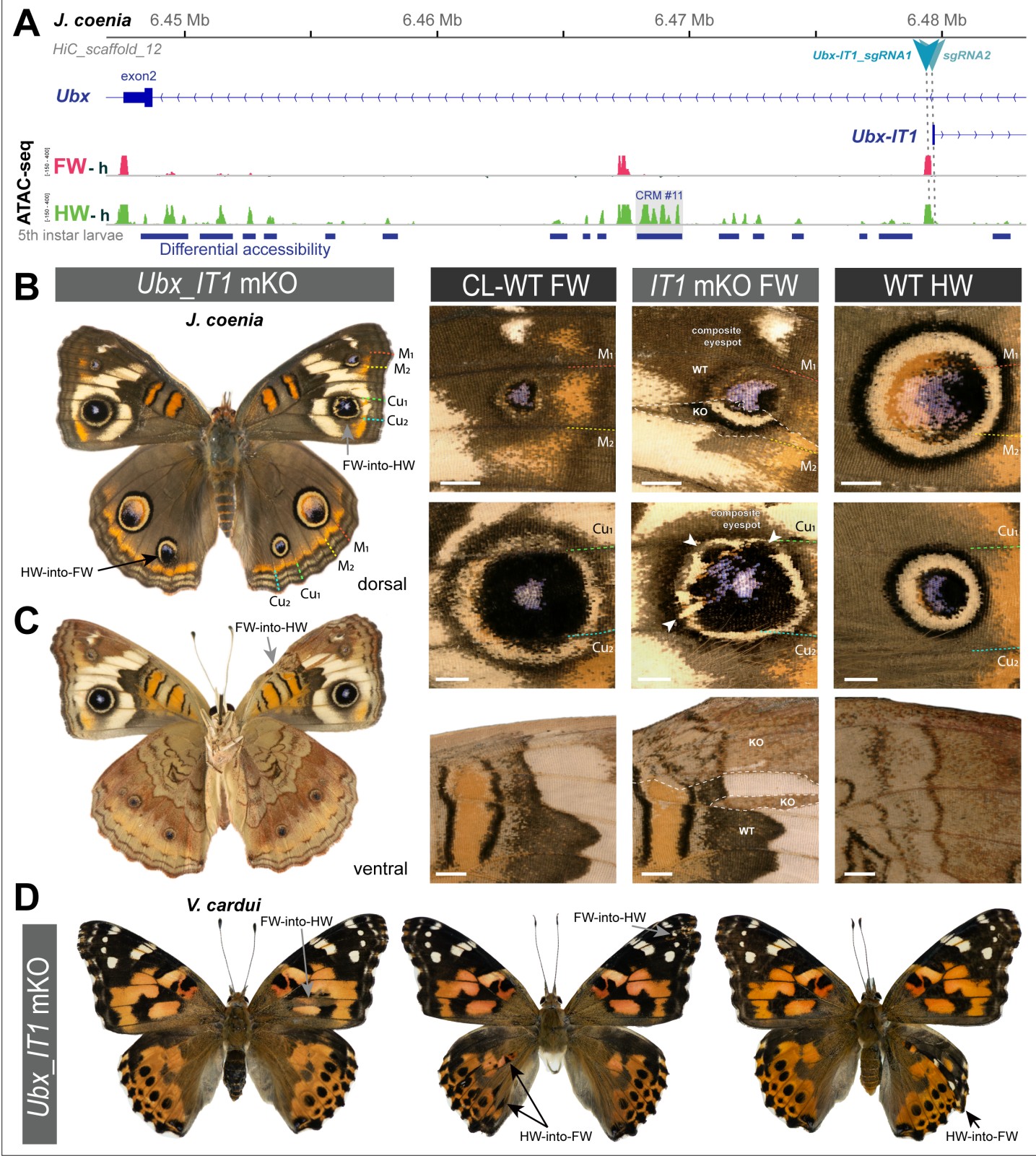

**Figure 5.** Rare, dual homeoses obtained from CRISPR mutagenesis of the *lncRNA_Ubx-IT1* 5' region. (**A**) Genomic context of the sgRNA targets (here shown in *J. coenia*), in the promoter and first exon of the non-coding *Ubx-IT1* transcript. (**B–C**) Dorsal and ventral views of a *J. coenia* crispant displaying dual homeoses, that is with both FW→HW (presumably due to *Ubx* gain-of-expression), and HW→FW clones (akin to *Ubx* null mutations). Insets on the right describe forewing mutant clones (*IT1 mKO*), in apposition to CL-WT (contralateral forewings from the same individual), and WT HW (wild type

*Figure 5 continued on next page*

*Figure 5 continued*

hindwings from the same individual and mutant forewing side). (**D**) Examples of dual homeoses obtained when targeting orthologous sites in *V. cardui*. Scale bars: 1 mm.

The online version of this article includes the following figure supplement(s) for figure 5:

**Figure supplement 1.** Additional mutant phenotypes from CRISPR-mediated interrogation of *lncRNA_Ubx-IT1* 5′ region in *J. coenia* (top) and *V. cardui* (bottom).

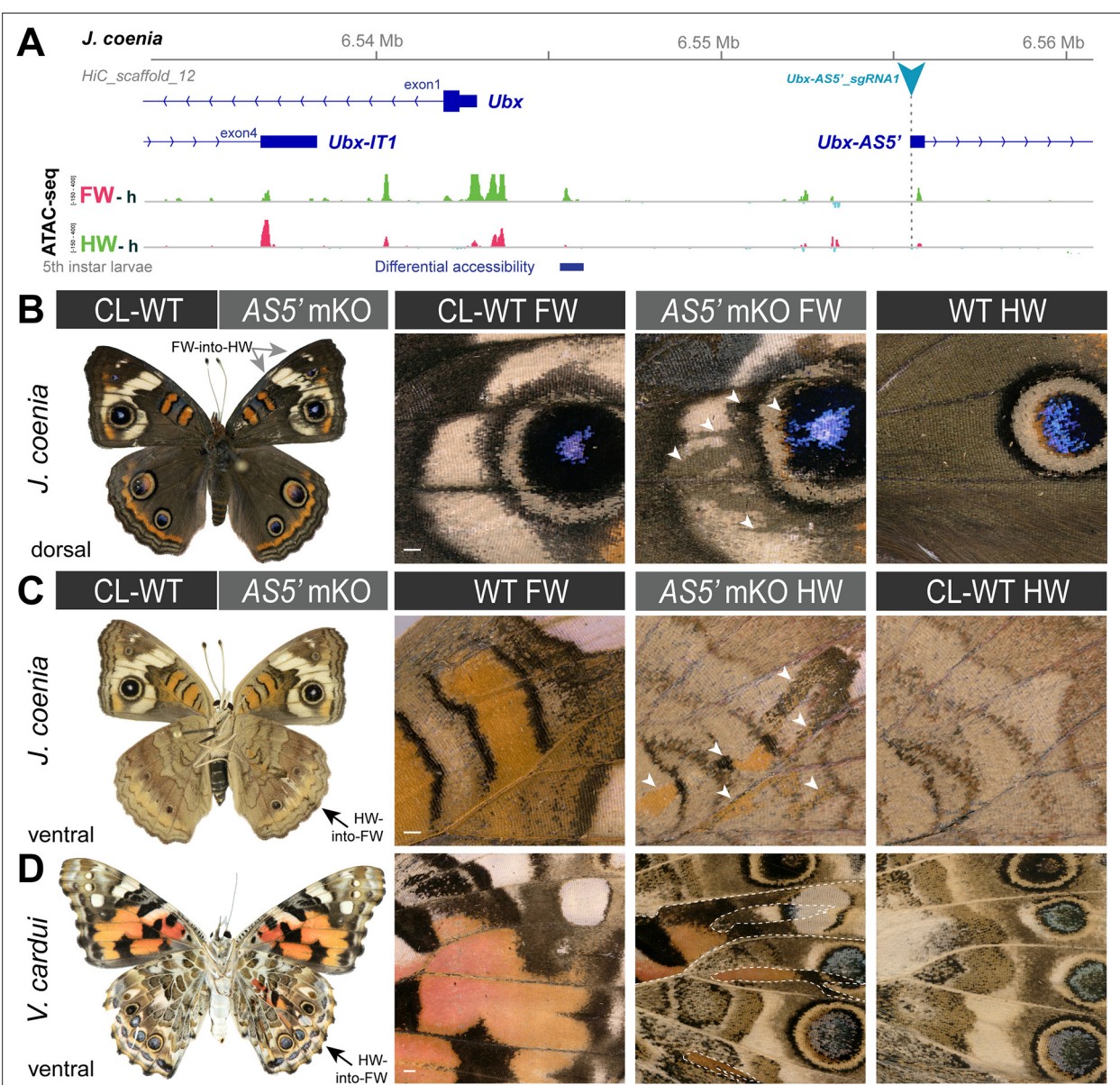

**Figure 6.** Homeoses obtained from CRISPR mutagenesis of the lncRNA *Ubx-AS5′* first exon. (**A**) CRISPR sgRNA targets (here shown in *J. coenia*), in the first exon of the non-coding *Ubx-AS5′* transcript. (**B**) A single *J. coenia* crispant showed a FW→HW transformation. Insets on the right describe forewing mutant clones (*AS5′ mKO*), in apposition to CL-WT (contralateral forewings from the same individual), and WT HW (wild-type hindwings from the same individual and mutant forewing side). (**C–D**) Examples of HW→FW homeoses obtained in *J. coenia* or when targeting orthologous sites in *V. cardui*. Scale bars: 500 μm.

The online version of this article includes the following figure supplement(s) for figure 6:

**Figure supplement 1.** Additional mutant phenotypes from CRISPR-mediated interrogation of the *lncRNA_Ubx-AS*5′ region in *J. coenia* and *V. cardui*.

mutants and a single forewing mutant (*Figure 6* and *Figure 6—figure supplement 1*). As with *Ubx-IT1* CRISPR experiments, these results may be explained by regulatory disruption of *Ubx* transcription, with a higher ratio of hindwing phenotypes compared to forewings linked to the proximity of the *Ubx* promoter. Overall, we conclude that the mutational interrogation at these loci can result in dual loss (hindwing) and gain (forewing) of *Ubx* function effects. Deciphering whether or when these effects affected *Ubx* expression via local *cis*-regulatory modules, impairment of lncRNA transcripts, or larger indels overlapping with *Ubx* exons, will require further study (see Discussion).

## Dual effects of mutagenesis in a putative *Ubx cis*-regulatory module

In an attempt to probe for *Ubx* hindwing-specific regulatory sequences, we focused on a~25 kb region in the first intron of *Ubx* that displays an ATAC-seq signature of hindwing enrichment in open-chromatin relative to forewings, hereafter dubbed *CRM11* (*Figure 7A*). We subdivided this differentially accessible region into four peaks (*11 a, b, c,* and *d*). Targeting the ATAC-seq peaks with multiple sgRNAs spanning sub-domains *11* a and *11* c (*UbxCRE11a2c5* in *V. cardui*, *11* a2a3c5c6 in *J. coenia*), or with a single target in *11* c (*UbxCRE11c5* in *V. cardui*) yielded dual homeoses: FW➝HW and HW➝FW (*Figure 7B–D* and *Figure 7—figure supplement 1*). Hindwing effects were reminiscent of *Ubx* protein coding knockouts (*Tendolkar et al., 2021*), indicating that these crispant alleles with a hindwing phenotype produce *Ubx* loss-of-function effects. Individuals with hindwing clones were 2.75 times more common than individuals with forewings in this dataset. Similarly to the lncRNA loci perturbation experiments, dual homeoses may indicate the presence of hindwing activators and forewing repressors in the *CRM11* region, with various CRISPR alleles producing a spectrum of indels and effects (see Discussion). It is noteworthy that while single-target experiments showed little lethality (55% hatching rate), dual or quadruple injection mixes resulted in low hatching rates of injected embryos (~10%). Multiple targeting thus appears to induce high-rates of embryonic lethality, possibly due to chromosomal damage (*Cullot et al., 2019*; *Zuccaro et al., 2020*). Dual targeting with *a2 +c5* also yielded partial HW➝FW homeoses in *V. cardui* under the form of ectopic white eyespot foci phenotypes (*Figure 7E*), as occasionally observed in *Ubx* null crispants (*Tendolkar et al., 2021*), seemingly due to hypomorphic or heterozygous allelic states.

Next, we focused on a single target shared between both *V. cardui* and *J. coenia* in the *11b* subdomain. A whole genome alignment between 23 lepidopteran species and 2 trichopteran species indicated that region *11b* is deeply conserved, suggesting important functional constraints on its sequence (*Figure 7—figure supplement 2A–B*). Mutagenesis of *11b* using a single guide RNA (*Ubx11b9*) yielded a relatively high hatching rate (mean = 51.8 %), indicating a rare occurrence of the deleterious mutational effects observed in multiple targeting (see above). Four *J. coenia* crispants and two *V. cardui* crispants were obtained, all exclusively showing hindwing phenotype devoid of forewing effects. HW➝FW homeoses included a variety of phenotypes all seen in *Ubx* CDS mutants (*Tendolkar et al., 2021*), including transformations of the orange Discalis elements and the white band in *J. coenia*, and partial shifts in eyespot identity (*Figure 7—figure supplement 2C*). Together the consistency in direction of transformations and the deep conservation of the *11b* region suggests it may encode an enhancer necessary for the transcriptional activation of *Ubx* in hindwings.

## A sample of spontaneous homeotic mutants in *Heliconius* butterflies

Homeotic shifts between forewings and hindwings can occur naturally in Lepidoptera, and have been documented as pattern aberrations in museum specimens (*Sibatani, 1980*; *Sibatani, 1983*). As a complement to CRISPR-induced homeoses, we document here a rich sample of forewing/hindwing homeotic mutants in the genus *Heliconius*, systematically collected by L. E. Gilbert between 1987 and 2016 in captive stocks at UT Austin, as well as in the wild. Across these 15 spontaneous mutants, 12 show HW➝FW clones, against 3 specimens with FW➝HW effects (*Figure 8*, *Figure 8—figure supplements 1–2*). Mutant clones in this dataset were always posterior to the $M_2$ vein. Only 2 mosaic phenotypes were found on a dorsal side, with the 13 others appearing ventrally. These homeotic mosaics show pattern shifts with complete fore/hindwing conversions of scale types, as seen for instance in the loss of gray wing coupling scales on posterior ventral forewings (*Figure 8A–B*), or conversely, in their acquisition in posterior hindwings (arrowheads in *Figure 8—figure supplements 1–2*). Homeoses also include noticeable local changes in wing shape, particularly in hindwings (asterisks in *Figure 8—figure supplements 1–2*). Taken together, these effects are akin to CRISPR-induced perturbations at

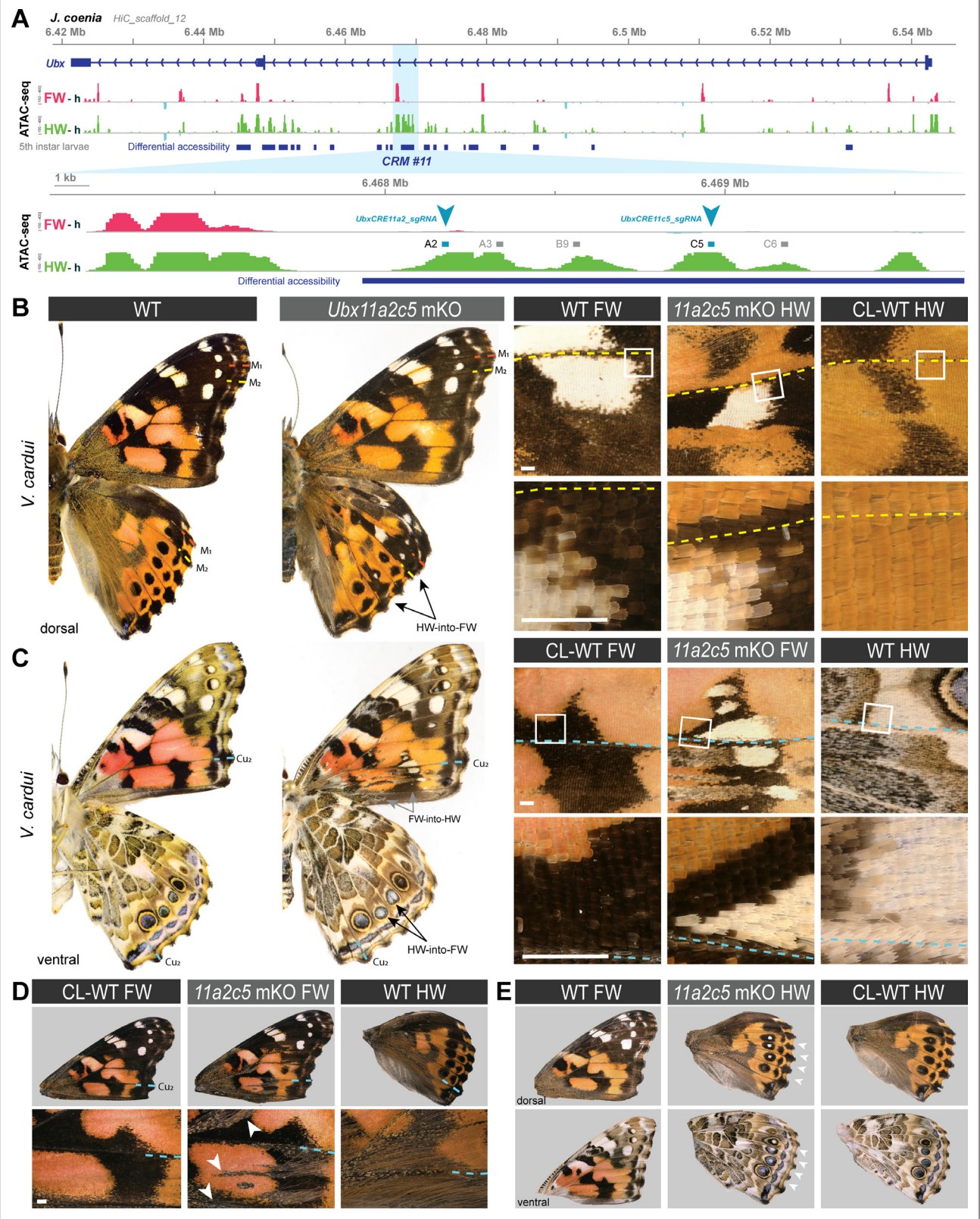

**Figure 7.** CRISPR perturbation of *Ubx CRM11* generates occasional dual homeotic phenotypes. (**A**) Overview of ATAC-seq differential chromatin accessibility profiles (hindwing - head tissues, green; forewing - head tissue, magenta) across the *Ubx* first exon. Several regions show differential opening between wings, one of which (*CRM11*), was targeted here for CRISPR perturbation (sites *a2* and *c5* indicate sgRNA targets). (**B**) Dual homeosis phenotypes obtained in *V. cardui* following dual-targeting of *UbxCRE11a2c5*, including homeoses in color patterns and scale morphology. (**D**) Additional

*Figure 7 continued on next page*

*Figure 7 continued*

example of a *V. cardui UbxCRE11a2c5* crispant with a forewing phenotype (gain of hindwing hair patches, arrowheads). (**E**) Example of mild hindwing homeoses showing a white eyespot focus on the dorsal and ventral sides. These effects were previously shown to occur in coding *Ubx* CRISPR knock-out experiments (*Tendolkar et al., 2021*). Contralateral (CL) WT wings are shown for comparison with mutant wings (**B–E**). Colored dashed lines: wing veins. Scale bars: 500 µm.

The online version of this article includes the following figure supplement(s) for figure 7:

**Figure supplement 1.** Additional mutant phenotypes from CRISPR-mediated interrogation of *CRM11* in *J. coenia* and *V. cardui* show bidirectional homeoses and non-homeotic eyespot changes.

**Figure supplement 2.** CRISPR perturbation of the conserved *Ubx_CRE11b* results in HW➞FW homeoses.

the *Ubx* locus. We speculate that fore/hindwing homeotic aberrations, found in nature and captive stocks, result from mutations at the *Ubx* locus itself.

## Discussion
### An intronic region with ATAC-seq hindwing-enrichment regulates *Ubx*

All CRISPR targets yielded homeotic phenotypes (*Figure 9*), with two kinds of interference with *Ubx* expression – forewing gain-of-function effects, and hindwing loss-of-function effects – and indicating the presence of regulatory sequences (broadly defined), that repress or enhance *Ubx* expression in this region. It is crucial here to highlight the limitations of the method, in order to derive proper insights about the functionality of the regulatory regions we tested. In essence, butterfly CRISPR experiments generate random mutations by non-homologous end joining repair, that are usually deletions (*Connahs et al., 2019*; *Mazo-Vargas et al., 2022*; *Van Belleghem et al., 2023*). Ideally, regulatory CRISPR-induced alleles should require genotyping in a second ($G_1$) generation to be properly matched to a phenotype (*Mazo-Vargas et al., 2022*). Possibly because of lethal effects, we failed to pass *Ubx* locus $G_0$ mutations to a $G_1$ generation for genotyping, and were thus limited here to mosaic analysis. As adult wings scales are dead cells, the genetic material building a given color phenotype is lost at this stage, but we circumvented this issue by genotyping a pupal forewing displaying an homeotic phenotype in the more efficient *Antp-Ubx_BE* perturbation experiment (*Figure 4—figure supplement 3*). In this case, PCR amplification of a 600 bp fragment followed by Sanger sequencing recovered signatures of indel variants, with mixed chromatograms starting at the targeted sites. But in all other experiments (*CRM11*, *IT1*, and *AS5'* targets), we did not genotype mutant tissues, as they were only detected in adult stages and generally with small clone sizes. Some of these clones may have been the results of large structural variants, as data from other organisms suggests that Cas9 nuclease targeting can generate larger than expected mutations that evade common genotyping techniques (*Shin et al., 2017*; *Adikusuma et al., 2018*; *Kosicki et al., 2018*; *Cullot et al., 2019*; *Owens et al., 2019*). Even under the assumption that such mutations are relatively rare in butterfly embryos, the fact we injected >100 embryos in each experiment makes their occurrence likely (*Figure 9*), and we are unable to assign a specific genotype to the homeotic effects we obtained in *CRM11*, *IT1* and *AS5'* perturbation assays.

When targeting hindwing-enriched ATAC-seq peaks within the first intron of *Ubx* – from *CRM11* to the hindwing-enriched open-chromatin peak that coincides with the first exon of *Ubx-IT1* – we obtained a mixture of hindwing and forewing phenotypes. Given the potential heterogeneity of allele sizes in these experiments, it is difficult to conclude robustly about the function of individual targets. Nonetheless, the phenotypic data and in particular the obtention of dual homeoses suggest we disrupted sequences that are necessary to *Ubx* activation in hindwings, as well as to its repression in forewings. Bifunctional *cis*-regulatory elements that can switch between enhancer and silencer roles are prevalent in *Drosophila* (*Gisselbrecht et al., 2020*; *Segert et al., 2021*; *Pang et al., 2023*). The *CRM11* and *IT1* targets adjoin or overlap with open-chromatin signals in both wing sets (*Figures 5A and 7A*), providing circumstantial evidence that these regions might serve as bifunctional elements. Similar observations were recently made in mutational interrogation experiments of the butterfly *WntA* patterning gene (*Mazo-Vargas et al., 2022*). Alternatively, silencers and enhancers may be tightly linked and interact in close proximity to shape gene expression (*Méndez-González et al.,*

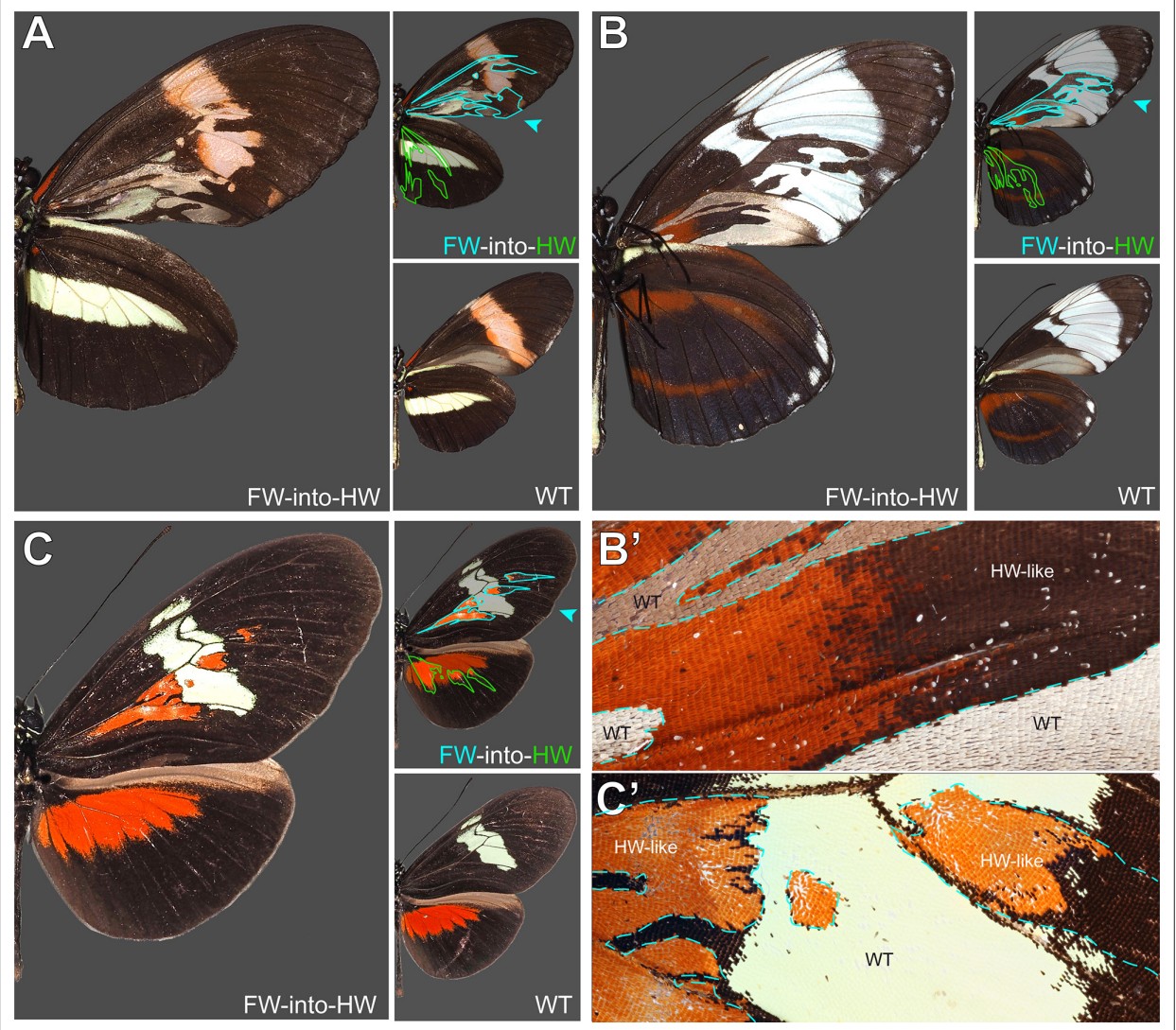

**Figure 8.** Mosaic forewing homeoses in *Heliconius* butterfly spontaneous mutants. Wild-type and mutant sides from the same individuals are shown in each panel, with one side digitally flipped to match left-to-right orientation. (**A**) *Heliconius melpomene rosina*, ventral view. Wild-caught in the Osa Peninsula (Costa Rica), October 1989. (**B**) *Heliconius cydno galanthus*, ventral view (magnified inset in **B'**). Stock culture from Organisation for Tropical Studies station, La Selva (Costa Rica), June 1990 (**C**) *Heliconius himera*, dorsal view (magnified inset in **C'**). Stock Culture in the butterfly farm Heliconius Butterfly Works in Mindo (Ecuador), March 2008.

The online version of this article includes the following figure supplement(s) for figure 8:

**Figure supplement 1.** Hindwing homeoses in *Heliconius* butterfly spontaneous mutants from pure stocks, hybrid cultures and wild-caught individuals from the L.

**Figure supplement 2.** Hindwing homeoses in *Heliconius* butterfly spontaneous mutants from pure stocks, hybrid cultures and wild-caught individuals from the L.

*2023*), implying in our case that forewing and hindwing phenotypes are mediated by alleles spanning adjacent but distinct elements. A formal test of these mechanisms would require germline transmission and genotyping of these alleles, which was unsuccessful in our attempts at crossing *Ubx* cis-regulatory crispants. In contrast with the dual effects obtained when targeting *CRM11a+c* (*Figure 9*), *CRM11b* perturbation resulted in hindwing-limited effects, and may suggest that an *Ubx* enhancer was consistently compromised in this specific dataset. The high lethality and small size of mutant wing streaks suggest that only individuals with sparse, small mutant mitotic clones can survive to the adult stage. If this is true, *CRM11* may contain pleiotropic enhancers that are vital for normal *Ubx* function at earlier stages, but expression-reporter studies will be required to test this.

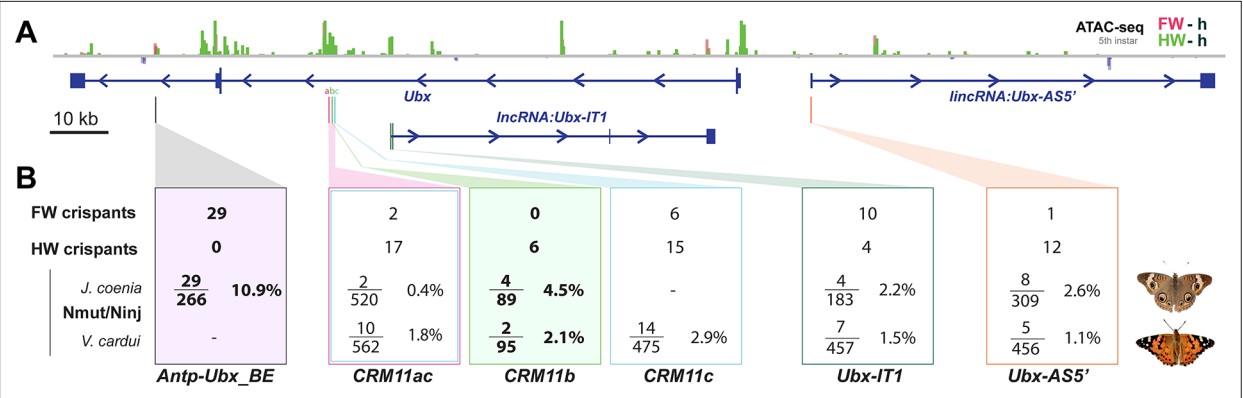

**Figure 9.** Summary of wing homeosis phenotypes obtained from mutational interrogation. (**A**) CRISPR targets at non-coding regions across the *Ubx* region, here visualized in *J. coenia*. (**B**) Summary of injection and adult phenotype data obtained across CRISPR experiments. FW/HW crispants: total number of individuals with forewing or hindwing homeotic clones, regardless of the injected species. Individuals with dual homeosis are counted in both categories. $N_{mut}/N_{inj}$: number of crispants obtained ($N_{mut}$), over the number of injected embryos for each species. Bold: experiments with consistent effects in only one segment. See **Table 1** for details.

## Parsing lncRNA-encoding regions – correlation or cause?

LncRNAs are emerging as important regulators of gene expression and developmental processes (*Zhang et al., 2019*; *Statello et al., 2021*). *IT1* targeting generated a majority of forewing phenotypes, suggesting perturbation of *Ubx* repression in the T2 segment. However, *IT1* showed low expression in forewing RNAseq datasets from *K. inachus,* and higher expression in the hindwing (*Figure 1B*), a pattern inconsistent with a repressive role of the antisense *IT1* transcript on *Ubx* expression. It is generally challenging to disentangle the effects of transcription of a non-coding element from the potential effects of adjacent enhancers (*Natoli and Andrau, 2012*; *Pease et al., 2013*). Therefore, an alternative explanation would be that the phenotypes are confounded by the overlap and proximity to open-chromatin regions, which may play *cis*-regulatory roles on *Ubx* via DNA-protein interactions, rather than via the lncRNA. If this is the case, it is possible that the targeted *Ubx-IT1* site, which yielded homeoses in both directions and bears both forewing and hindwing open-chromatin, is a bifunctional *cis*-regulatory element that can shift regulatory activities between these tissues (*Gisselbrecht et al., 2020*). Targeted mutagenesis of the *Ubx-AS5'* first exon mainly generated hindwing phenotypes, although with a relatively low-efficiency. Because this target is about 10 kb away from the *Ubx* promoter itself, it is plausible that the observed phenotypes were due to large deletions reaching the promoter region of *Ubx*. As mutational interrogation alone cannot discern if phenotypic effects arose from regulatory failure at the chromatin or transcript level, determining if *AS5'* and *IT1* are functional lncRNAs will need further examination. Of note, a systematic in-situ hybridization survey (*Pease et al., 2013*) showed that *Drosophila* embryos express an antisense transcript in its 5' region (*lncRNA:bxd*), as well as within its first intron (*lncRNA:PS4*). It is thought that *Drosophila bxd* regulates *Ubx*, possibly by transcriptional interference or by facilitation of the *Fub-1* boundary effect (*Petruk et al., 2006*; *Ibragimov et al., 2023*), while the possible regulatory roles of *PS4* remain debated (*Hermann et al., 2022*). While these dipteran non-coding transcripts lack detectable sequence similarity with the lepidopteran *IT1* and *AS5'* transcripts, further comparative genomics analyses of the *Ubx* region across the holometabolan insect phylogeny should clarify the extent to which Hox cluster lncRNAs have been conserved or independently evolved.

## A TAD boundary element likely acts as an insulator preventing *Ubx* forewing expression

Tight maintenance of TAD boundaries at the *Hox* locus is crucial for accurate segment identity and is facilitated by insulator proteins (*Stadler et al., 2017*; *Gambetta and Furlong, 2018*; *Ramírez et al., 2018*). The *Antp-Ubx_BE* element we targeted is in a good position to block interactions between *Antp* and *Ubx* (*Figures 2–3*). Consistent with this idea, the last intron of *Ubx* contains 8 CTCF binding motifs that are all clustered within 5 kb around the forewing-enriched ATAC peak, including two sites at highly conserved positions that are only 100 bp away from the CRISPR target

**Table 1.** CRISPR mutational interrogation experiments at putative *Ubx* regulatory regions.

| Species | sgRNA(s) | Inj. Embryos Ninj | L1 larvae Nhat | Pupae or L5 larvae | Adults Nadu | Crispants Nmut | Inj. time h AEL | Cas9:sgRNA ng/µL | Hatching Rate Nhat/Ninj | Crispant Rate Nmut/Ninj |
|---|---|---|---|---|---|---|---|---|---|---|
| | | 59 | 50 | 50 | 44 | 6 | 2.5–3.5 | 500 : 250 | 84.7% | 10.2% |
| | | 118 | 40 | 40 | 31 | 6 | 1.75–2.75 | 250 : 125 | 33.9% | 5.1% |
| | *Antp-Ubx_BE* | 89 | 44 | *44* | *39* * | 17 | 2.25–3.5 | 500 : 250 | 49.4% | 19.1% |
| *J. coenia* | Total | 266 | 90 | 134 | 115 | 29 | | | 33.8% | 10.9% |
| | | 204 | 67 | 50 | 50 | 2 | 1–3 | 250 : 125 | 32.8% | 1.0% |
| | | 108 | 49 | 31 | 31 | 3 | 2–3 | 125 : 62.5 | 45.4% | 2.8% |
| | *IT1_sgRNA1* | 145 | 60 | 39 | 39 | 2 | 2.25–3.5 | 500 : 250 | 41.4% | 1.4% |
| *V. cardui* | Total | 457 | 176 | 120 | 120 | 7 | | | 38.5% | 1.5% |
| | | 59 | 40 | 7 | 6 | 0 | 0.5–2.5 | 500 : 250 | 67.8% | 0.0% |
| | *IT1_sgRNA2* | 124 | 112 | 112 | 110 | 4 | 2.25–4.75 | 500 : 250 | 90.3% | 3.2% |
| *J. coenia* | Total | 183 | 152 | 119 | 116 | 4 | | | 83.1% | 2.2% |
| | | 334 | 183 | 57 | 52 | 5 | 2–3 | 250 : 125 | 54.8% | 1.5% |
| | *AS5_sgRNA1* | 122 | 87 | 2 | 2 | 0 | 2–4 | 500 : 250 | 71.3% | 0.0% |
| *V. cardui* | Total | 456 | 270 | 59 | 54 | 5 | | | 59.2% | 1.1% |
| *J. coenia* | *AS5_sgRNA1* | 309 | 181 | 181 | 181 | 8 | 2–4.5 | 500 : 250 | 58.6% | 2.6% |
| | *Ubx11a2+3 +c5+6* | 317 | 18 | - | - | 2 | 1–3 | 500 : 75 ea. | 5.7% | 0.6% |
| | | 203 | 35 | 0 | 0 | 0 | 1.5–3.5 | 500 : 75 ea. | 17.2% | 0.0% |
| *J. coenia* | Total | 520 | 53 | - | - | 2 | | | 10.2% | 0.4% |
| | | 50 | 5 | 3 | 3 | 2 | 4–4.5 | 500 : 500 | 10.0% | 4.0% |
| | | 151 | 29 | 6 | 5 | 2 | 2–2.75 | 500:125:125 | 19.2% | 1.3% |
| | *Ubx11a2+c5* | 361 | 18 | 13 | 16 | 6 | 0.5–2 | 500:125:125 | 5.0% | 1.7% |
| *V. cardui* | Total | 562 | 52 | 22 | 24 | 10 | | | 9.3% | 1.8% |
| | | 168 | 99 | 27 | 26 | 3 | 3.75–4.75 | 250 : 125 | 58.9% | 1.8% |
| | | 62 | 22 | 9 | 9 | 2 | 0.5–0.75 | 500 : 250 | 35.5% | 3.2% |
| | | 131 | 93 | 8 | 8 | 3 | 1.5–3 | 500 : 250 | 71.0% | 2.3% |
| | *Ubx11c5* | 114 | 63 | 20 | 20 | 6 | 3.5–4.5 | 500 : 250 | 55.3% | 5.3% |
| *V. cardui* | Total | 475 | 277 | 64 | 63 | 14 | | | 58.3% | 2.9% |
| | | 32 | 18 | 6 | 5 | 1 | 1.25–2.25 | 500 : 250 | 56.3% | 3.1% |
| | *Ubx11b9* | 63 | 49 | 9 | 6 | 1 | 3.5–4.5 | 500 : 250 | 77.8% | 1.6% |
| *V. cardui* | Total | 95 | 67 | 15 | 11 | 2 | | | 70.5% | 2.1% |
| | | 41 | 13 | 13 | 13 | 3 | 2.5–4 | 125 : 62.5 | 31.7% | 7.3% |
| | *Ubx11b9* | 48 | 21 | 14 | 14 | 1 | 2–3 | 250 : 125 | 43.8% | 2.1% |
| *J. coenia* | Total | 89 | 34 | 27 | 27 | 4 | | | 38.2% | 4.5% |

*: upper estimate, includes 16 fifth instars larvae that were dissected for immunostainings, of which 7 were mutants (as evidenced by ectopic UbdA in forewings), and 3 dissected mutant pupae

(*Figure 2—figure supplement 1*). CTCF sites prevent cross-talk between regulatory domains in the fly BX-C complex, and result in *Hox* misexpression when deleted (*Postika et al., 2018*; *Kyrchanova et al., 2020*; *Kaushal et al., 2022*; *Kahn et al., 2023*). Thus, the density of CTCF sites in this region may be indicative of a *bona fide* insulator active in forewings. CRISPR mutagenesis of *Antp-Ubx_BE* generated FW→HW homeoses associated with a gain of UbdA antigenicity in forewings, with no effects in the other direction, in stark contrast with other targets (*Figure 9B*). This suggests a possible loss of the TAD boundary in the crispant clones, resulting in a TAD fusion or in a long-range inter-action between a T2-specific enhancer and *Ubx* promoter. Similar deletion alleles resulting in a TAD fusion and misexpression effect have been described at the *Notch* locus in *Drosophila* (*Arzate-Mejía et al., 2020*), in digit-patterning mutants in mice and humans (*Lupiáñez et al., 2015*; *Anania et al., 2022*), or at murine and fly *Hox* loci depleted of CTCF-mediated regulatory blocking (*Narendra et al., 2015*; *Gambetta and Furlong, 2018*; *Kyrchanova et al., 2020*). It will be interesting to profile the binding of insulator proteins and transcriptional repressors across the butterfly *Hox* TAD landscape to shed more light onto the mechanisms of *Ubx* insulation, using in vivo assays (*Bowman et al., 2014*), or in silico predictions that take advantage of updated binding matrices for insect insulator proteins (*Mitra et al., 2018*). Of note, our CRISPR target is adjacent to an hindwing-enriched peak that also presented CTCF binding sites (*Figure 4A*). Following a similar logic, this site could be a candidate insulator specific to *Ubx*-expressing tissues like the hindwing, a hypothesis that will require further testing. Lastly, it is worth noting that the *Antp/Ubx* TAD boundary we identified is intragenic, within the last intron of *Ubx*. It is unclear if this feature affects transcription, but this configuration might be analogue to the *Notch* locus in *Drosophila*, which includes a functional TAD boundary in an intronic position (*Arzate-Mejía et al., 2020*).

## Making sense of spontaneous wing homeotic mutants

In this article, we documented a large sample of spontaneous homeotic mutants obtained in *Heliconius spp.* All homeotic clones were limited to the wing posterior compartments (i.e. posterior to the $M_2$ vein), possibly because of parasegmental, compartment-specific regulatory domains that played historic roles in the study of *Drosophila* BX-C regulation (*Maeda and Karch, 2015*). Sibatani documented in Lepidoptera that "*the mosaics of F/H homeosis tend to occur most frequently in the posterior half of the wing, the boundary of the anterior and posterior halves occurring some-where in space $M_1$-$M_2$*" (*Sibatani, 1983*). Our collection of spontaneous *Heliconius* mutants only displayed clones in posterior regions, consistently with this trend. However, our CRISPR perturba-tion assays of *J. coenia* and *V. cardui cis*-regulatory regions also generated anterior clones, with all targets. Deciphering how butterfly *Ubx* regulation is compartmentized between parasegmental or wing antero-posterior domains will require additional investigation. Most *Heliconius* homeoses were in the hindwings (i.e. putative *Ubx* loss-of-expression clones), and among these, all but one were ventral (*Figure 4—figure supplements 1–2*). Three mutants showed forewing homeoses (i.e. putative *Ubx* gain-of-expression clones), two of them ventral and one of them dorsal (*Figure 8*). The systematic reviews of wing homeosis in Lepidoptera found that forewing homeoses are almost as common as hindwing ones (*Sibatani, 1980*; *Sibatani, 1983*). Our mutational interrogation assays, while coarse in nature, revealed the existence of activating and repressing *cis*-regulatory sequences at the *Ubx* locus itself. Spontaneous FW↔HW homeoses observed in butterflies and moths may thus result from somatic mutations or transposition events at this locus.

## Materials and methods
### Genome annotations and transcriptomic analysis

Nymphalid genome sequences of the *Hox* cluster and their annotations were extracted from the NCBI Assembly and Lepbase online repositories (*Challi et al., 2016*; *Kitts et al., 2016*) as follows: *V. cardui* from NCBI *ilVanCard2.1* and LepBase *Vc_v1*; *A.* (*Nymphalis*) *io* from NCBI *ilAgIloxx1.1*; *J. coenia* from Lepbase *Jc_v2*; *P xylostella* from NCBI *ilPluXylo3.1*. The *Ubx* regions from *ilVanCard2.2, Vc_v1, and Jc_v2* were manually re-annotated using wing transcriptome data on the NCBI SRA (BioProj-ects *PRJNA661999, PRJNA293289, PRJNA237755, PRJNA385867, and PRJNA498283*) The genome sequence of *K. inachus* was obtained from the Dryad repository (*Yang et al., 2020*). Differential gene expression analysis across the *K. inachus Ubx* locus were carried out using wing transcriptome data

available on the NCBI SRA (BioProject *PRJNA698433*), following a manual re-annotation of a preliminary gene models provided by Peiwen Yang and Wei Zhang (*Wang et al., 2022*). All transcripts analyses were performed using the *STAR* intron-aware aligner and *DEseq2* expression analysis package as previously described (*Love et al., 2014*; *Dobin and Gingeras, 2016*; *Hanly et al., 2019*; *Hanly et al., 2022*). Expression levels were calculated as genome-wide normalized counts and pairwise Wald tests were performed to assess differential expression between forewings and hindwings. Multiple test adjustment was performed using Benjamini and Hochberg correction.

## Hi-C and ATAC-seq analyses

Hi-C data from *J. coenia* fifth instar larval forewings and 48–72 hr APF pupal hindwings are available at the NCBI SRA BioProject *PRJNA641138* (*van der Burg et al., 2020*). Triplicated ATAC-seq datasets for larval and pupal wing and head tissues of *J. coenia* and *V. cardui* (*van der Burg et al., 2019*; *Mazo-Vargas et al., 2022*) are available on the NCBI SRA BioProjects *PRJNA497878*, *PRJNA695303*, and *PRJNA559165*. All the ATAC-seq and Hi-C data were re-analysed on *J. coenia* and *V. cardui Ubx* genomic regions as previously described (*Mazo-Vargas et al., 2022*). Briefly, matrices of interactions were constructed by mapping paired reads against the *Junonia coenia* genome (*Mazo-Vargas et al., 2022*) using *hicBuildMatrix* (*Ramírez et al., 2018*). Matrices from larvae and pupae were normalized using *hicNormalize* and corrected with the Knight-Ruiz matrix balancing algorithm. The definitions of topologically associating domains (TADs) can be influenced by various factors such as the choice of software, parameters, sequencing depth, and the presence of experimental noise. To ensure reliability, it is recommended to compare TAD calls with independent datasets, such as histone marks or known factors associated with TAD boundaries. In the absence of these specific datasets, we employed a different combination of parameters in the *hicFindTADs* tool and compared the resulting TAD calls. HiC matrices at 10 kb and 20 kb bin resolutions were utilized, and TAD insulation scores were evaluated using a false-discovery rate correction for multiple testing, with *p-value* thresholds of 0.01 and 0.005. Consistent TAD boundary calls with negative TAD separation scores were selected to define domain limits at 10 kb and 20 kb matrix resolutions.

## CTCF motif binding predictions

The program *fimo* was used to scan for the *J. coenia* candidate TAD boundary region (HiC_scaffold_12:6430000–6444000) for canonical CTCF binding sites, using the positional weight matrix MA0205.1 deposited in the JASPAR database (*Holohan et al., 2007*; *Cuellar-Partida et al., 2012*; *Castro-Mondragon et al., 2022*).

## Genomic conservation analyses

We generated whole-genome alignments of 25 Lepidoptera and 2 Trichoptera reference species from NCBI Assembly using *ProgressiveCactus* (*Armstrong et al., 2020*), and *HAL tools* (*Hickey et al., 2013*) for converting the resulting HAL file to the MAF format. We provided a species topology tree of 23 Lepidoptera species to *PhyloFit* (*Hubisz et al., 2011*) to fit a multiple sequence alignment on the reference *J. coenia Ubx* locus, using *HKY85* as the substitution model. Using *PhastCons* (*Siepel et al., 2005*), we then generated conservation score plots using standard parameters (target-coverage=0.45; expected-length=12; rho = 0.1) stored in BED and WIG file formats.

## Butterfly rearing and CRISPR microinjections

*J. coenia* and *V. cardui* colonies were maintained at 25 °C and 60–70% relative humidity in a growth chamber with a 14:10 light:dark photoperiod. Larval rearing on artificial diets, egg collection, and microinjections followed previously described methods (*Martin et al., 2020*; *Tendolkar et al., 2021*). Cas9:sgRNA heteroduplexes were prepared as previously described (*Martin et al., 2020*). Frozen aliquots of Cas9-2xNLS (2.5 µL; 1000 ng/µL) and sgRNA (2.5 µL; 500 ng/µL) were mixed in 2:1 and 4:1:1 mass ratios for single and dual target injections, respectively. CRISPR sgRNA targets are listed in *Supplementary file 1*.

## Genotyping

For verification in DNA lesions at the intended Antp-Ubx_BE site, a pupal wing fragment harboring visible mutant clones (*Figure 4—figure supplement 3B"*) or control wild-type tissue were PCR

amplified using the diluted protocol of the Phire Animal Tissue Direct PCR Kit (ThermoFisher) and a pair of oligonucleotides (Forward: 5'-ACCGATCGTAAACGTCAACTTAACG-3'; Reverse 5'-TACT GCGGTGGCGAGTGAATG-3'), before purification and Sanger sequencing using the reverse primer.

## Antibody stainings

Fifth instar wing disks were dissected in ice cold Phosphate Buffer Saline (PBS), fixed for 15–20 min at room temperature in methanol-free formaldehyde diluted to 4% in PBS / 2 mM EGTA (egtazic acid), washed in PBS with 0.1% Triton X-100 (PT), stored in PT with 0.5% Bovine Serum albumin (PT-BSA), incubated overnight at 4 °C in PT-BSA with a 1:5 dilution of the anti-UbdA peptide antibody serum (mouse monoclonal FP6.87, Developmental Studies Hybridoma Bank), and washed in PT. A 1:250 dilution of anti-Mouse IgG antibody coupled to AlexaFluor488 or Rabbit AlexaFluor555 was made in PT-BSA and spun down at 14,000 rcf to remove aggregates, and incubated with wings for 2 h at room temperature, before additional washes, incubation in 50% glycerol-PBS with DAPI (4',6-diamidino-2-phenylindole) nuclear stain, and incubation and mounting in 60% glycerol-PBS with 2 mM of EDTA (Ethylenediaminetetraacetic acid).

## Imaging

Full-mount photographs of *J. coenia* and *V. cardui* were taken on a Nikon D5300 digital camera mounted with an AF-S VR MicroNikkor 105 mm f/2.8 G lens, with magnified views taken using a Keyence VHX-5000 digital microscope mounted with VH-Z00T and VH-Z100T lenses. Immunofluorescent stainings were imaged on an Olympus BX53 epifluorescent microscope mounted with UPLFLN 4 x/0.13 and 10 X/0.3 objectives.

## Acknowledgements

We thank Ling Sheng Loh and the undergraduate researchers from the Martin Lab for assistance with micro-injections and insect rearing, Rachel Canalichio and the GWU Harlan Greenhouse personnel for growing host plants, Patricia Hernandez for sharing microscopes, and Alex Wild for assistance with *Heliconius* microphotographs at UT Austin. We wish to acknowledge James Lewis and Bob Reed for stimulating insights on open-chromatin biology and the *Hox* locus, as well as for generating Hi-C libraries published in previous publications that we re-analyzed here. This work was supported by the NSF awards IOS-1656553 and IOS-2110534 to AM, the Wilbur V Harlan Research Fellowship to AT, the NSF Postdoctoral Research Fellowship in Biology 2109536 to AMV, and the Smithsonian Institution Biodiversity Genomics Fellowship to JJH.

## Additional information

### Funding

| Funder | Grant reference number | Author |
|---|---|---|
| National Science Foundation | IOS-1656553 | Arnaud Martin |
| National Science Foundation | IOS-2110534 | Arnaud Martin |
| George Washington University | Wilbur V Harlan Research Fellowship | Amruta Tendolkar |
| National Science Foundation | 2109536 | Anyi Mazo-Vargas |
| Smithsonian Institution | Biodiversity Genomics Fellowship | Joseph J Hanly |

The funders had no role in study design, data collection and interpretation, or the decision to submit the work for publication.

## Author contributions
Amruta Tendolkar, Formal analysis, Investigation, Writing - original draft; Anyi Mazo-Vargas, Formal analysis, Investigation, Writing - review and editing; Luca Livraghi, Investigation, Writing - review and editing; Joseph J Hanly, Formal analysis, Writing - review and editing; Kelsey C Van Horne, Lawrence E Gilbert, Investigation; Arnaud Martin, Supervision, Writing - original draft

## Author ORCIDs
Luca Livraghi ⓘ http://orcid.org/0000-0002-2597-7550
Joseph J Hanly ⓘ http://orcid.org/0000-0002-9459-9776
Arnaud Martin ⓘ http://orcid.org/0000-0002-5980-2249

Joint Public Review: https://doi.org/10.7554/eLife.90846.3.sa1
Author Response https://doi.org/10.7554/eLife.90846.3.sa2

# Additional files

## Supplementary files
• Supplementary file 1. List of sgRNAs used in CRISPR experiments.
• MDAR checklist

## Data availability
The current manuscript used previously published data that are refererenced in the Materials and methods section. No genomic data have been generated for this manuscript.

The following previously published datasets were used:

| Author(s) | Year | Dataset title | Dataset URL | Database and Identifier |
|---|---|---|---|---|
| Zhang L | 2021 | Vanessa cardui Genome sequencing and assembly | https://www.ncbi.nlm.nih.gov/bioproject/PRJNA661999 | NCBI BioProject, PRJNA661999 |
| Reed RD | 2015 | Vanessa cardui Raw sequence reads | https://www.ncbi.nlm.nih.gov/bioproject/PRJNA293289 | NCBI BioProject, PRJNA293289 |
| Daniels EV | 2014 | De novo transcriptome analysis profiles gene expression underlying seasonal polyphenism in butterfly wing patterns | https://www.ncbi.nlm.nih.gov/bioproject/PRJNA237755 | NCBI BioProject, PRJNA237755 |
| Zhang L | 2017 | A single master regulatory gene optix underlies both color and iridescence in butterflies | https://www.ncbi.nlm.nih.gov/bioproject/PRJNA385867 | NCBI BioProject, PRJNA385867 |
| van der Burg KRL | 2018 | Contrasting roles of transcription factors spineless and EcR in the highly dynamic chromatin landscape of butterfly wing metamorphosis (buckeye) | https://www.ncbi.nlm.nih.gov/bioproject/PRJNA498283 | NCBI BioProject, PRJNA498283 |
| Wang S, Teng D, Li X, Yang P, Da W, Zhang Y, Zhang Y, Liu G, Zhang X, Wan W, Dong Z, Wang D, Huang S, Jiang Z, Wang Q, Lohman DJ, Wu Y, Zhang L, Jia F, Westerman E, Zhang L, Wang W, Zhang W | 2021 | The genetics of leaf mimicry in Kallima inachus | https://www.ncbi.nlm.nih.gov/bioproject/PRJNA698433 | NCBI BioProject, PRJNA698433 |

*Continued*

| Author(s) | Year | Dataset title | Dataset URL | Database and Identifier |
|---|---|---|---|---|
| van der Burg KRL | 2020 | Genomic architecture and evolution of a seasonal reaction norm [Hi-C] (buckeye) | https://www.ncbi.nlm.nih.gov/bioproject/PRJNA641138 | NCBI BioProject, PRJNA641138 |
| van der Burg KRL | 2018 | Contrasting roles of transcription factors spineless and EcR in the highly dynamic chromatin landscape of butterfly wing metamorphosis (buckeye) | https://www.ncbi.nlm.nih.gov/bioproject/?term=PRJNA497878 | NCBI BioProject, PRJNA497878 |
| Lewis JJ | 2021 | Chromatin landscape of butterfly developing wings | https://www.ncbi.nlm.nih.gov/bioproject/PRJNA695303 | NCBI BioProject, PRJNA695303 |
| van der Burg KRL | 2020 | Genomic architecture and evolution of a seasonal reaction norm [ATAC-seq] (buckeye) | https://www.ncbi.nlm.nih.gov/bioproject/PRJNA559165 | NCBI BioProject, PRJNA559165 |

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
