## [Editor Report · eLife assessment]

This **valuable** paper examines the Bithorax complex in several butterfly species, in which the complex is contiguous and not split, as it is in the well-studied fruit fly *Drosophila*. Based on genetic screens and genetic manipulations of a boundary element involved in segment-specific regulation of Ubx, the authors provide **convincing** evidence for their conclusions, which could be strengthened by additional data and analyses in the future. The data presented are relevant for those interested in the evolution and function of Hox genes and of gene regulation in general.

---

## [Referee Report · Joint Public Review]

Summary:

The existence of hox gene complexes conserved in animals with bilateral symmetry and in which the genes are arranged along the chromosome in the same order as the structures they specify along the anteroposterior axis of organisms is one of the most spectacular discoveries of recent developmental biology. In brief, homeotic mutations leads to the transformation of a given body segment of the fly into the copy of the next adjacent segment. For the sake of understanding the main observation of this work, it is important to know that in loss-of-function (LOF) alleles, a given segment develops like a copy of the segment immediately anterior to it, and in gain-of-function mutations (GOF), the affected segment develop like a copy of the immediately posterior segment. Over the last 30 years the molecular lesions associated with GOF alleles led to a model where the sequential activation of the hox genes along the chromosome result from the sequential opening of chromosomal domains. Most of these GOF alleles turned out to be deletions of boundary elements (BE) that define the extend of the segment-specific regulatory domains. The fruit fly *Drosophila* is a highly specialized insect with a very rapid mode of segmentation. Furthermore, the hox clusters in this lineage have split. Given these specificities it is legitimate to question whether the regulatory landscape of the BX-C we know of in *D. melanogaster* is the result of very high specialization in this lineage, or whether it reflects a more ancestral organization. In this article, the authors address this question by analyzing the continuous hox cluster in butterflies. They focus on the integenic region between the Antennapedia and the Ubx gene, where the split occurred in *D. melanogaster*. Hi-C and ATAC-seq data suggest the existence of a boundary element between 2 Topologically-Associated-Domain (TAD) which is also characterized by the presence of CTCF binding sites. Butterflies have 2 pairs of wings originating form T2 (forewing) specified by Antp and T3 specified by Ubx (hindwing). Remarkably, CRISPR mutational perturbation of this boundary leads to the hatching of butterflies with homeotic clones of cells with hindwings identities in the forewing (a posteriorly oriented homeotic transformation). In agreement with this phenotype, the authors observe ectopic expression of Ubx in these clones of cells. In other words, CRISPR mutagenesis of this BE region identified by molecular tool give rise to homeotic transformations directed towards more posterior segment as the boundary mutations that had been 1st identified on the basis of their posterior oriented homeotic transformation in *Drosophila*. None of the mutant clones they observed affect the hindwing, indicating that their scheme did not affect the nearby Ubx transcription unit. This is a reassuring and important 1st evidence that some of the regulatory paradigm that have been proposed in fruit flies are also at work in the common ancestor to *Drosophilae* and Lepideptora.

Given the large size of the Ubx transcription unit and its associated regulatory regions it is not surprising that the authors have identified ncRNA that are conserved in 4 species of Nymphalinae butterflies, some of which also present in

Strengths: the convincing GOF phenotype resulting from the targeting of the Antp-Ubx_BE

Weaknesses: the lack of comparisons with the equivalent phenotypes obtained in *D. melanogaster* with for example the Fub mutation

---

## [Author Response]

The following is the authors’ response to the original reviews.

**eLife assessment**
This valuable paper examines the Bithorax complex in several butterfly species, in which the complex is contiguous and not split, as it is in the well-studied fruit fly *Drosophila*. Based on genetic screens and genetic manipulations of a boundary element involved in segment-specific regulation of Ubx, the authors provide solid evidence for their conclusions, which could be further strengthened by additional data and analyses. The data presented are relevant for those interested in the evolution and function of Hox genes and of gene regulation in general.

We are deeply grateful to the eLife editorial team and the two reviewers for their thoughtful and constructive feedback. We have used this feedback to improve our manuscript and have provided a point-by-point response below.

**Public Reviews:**

**Reviewer #1 (Public Review):**
In their article, "Cis-regulatory modes of Ultrabithorax inactivation in butterfly forewings," Tendolkar and colleagues explore Ubx regulation in butterflies. The authors investigated how Ubx expression is restricted to the hindwing in butterflies through a series of genomic analyses and genetic perturbations. The authors provide evidence that a Topologically Associated Domain (TAD) maintains a hindwing-enriched profile of chromatin around Ubx, largely through an apparent boundary element. CRISPR mutations of this boundary element led to ectopic Ubx expression in forewings, resulting in homeotic transformation in the wings. The authors also explore the results of the mutation in two non-coding RNA regions as well as a possible enhancer module. Each of these induces homeotic phenotypes. Finally, the authors describe a number of homeotic phenotypes in butterflies, which they relate to their work.Together, this was an interesting paper with compelling initial data. That said, I have several items that I feel would warrant further discussion, presentation, or data.First, I would not state, "Little is known about how Hox genes are regulated outside of flies." They should add "in insects" since so much in known in vertebrates

Corrected

For Figure 1, it would aid the readers if the authors could show the number of RNAseq reads across the locus. This would allow the readership to evaluate the frequency of the lncRNAs, splice variants, etc.

We have found it useful in the past to feature “Sashimi Plots”, as they provide a good overview of transcript splicing junctions and read support. Here we could not accommodate this in our Fig. 1A as this would require compiling the RNAseq reads from many tissues and stages to be meaningful, and we would lose the resolution on forewing vs hindwing tissues that is important in this article (only the Kallima inachus dataset allows this comparison, and was used in Fig 1B). More specifically, the wing transcriptomes available for J. coenia and V. cardui are not deep enough to provide a good visualization of Antp alternative promoter usage or on AS5’ transcription.

How common are boundary elements within introns? Typically, boundary elements are outside gene bodies, so this could be explored further. This seems like an interesting bit of biology which, following from the above point, it would be interesting to, at a minimum, discuss, but also relate to how transcription occurs through a possible boundary element (are there splice variants, for example?).

We do not see evidence of alternative splicing, and prefer to avoid speculating on transcriptional effects, but we agree that the intragenicity of the TAD boundary is interesting. We briefly highlighted this point in the revised Discussion:

"Lastly, it is worth noting that the Antp/Ubx TAD boundary we identified is intragenic, within the last intron of Ubx. It is unclear if this feature affects Ubx transcription, but this configuration might be analogue to the Notch locus in *Drosophila*, which includes a functional TAD boundary in an intronic position (Arzate-Mejía et al. 2020)."

The CRISPR experiments led to compelling phenotypes. However, as a *Drosophila* biologist, I found it hard to interpret the data from mosaic experiments. For example, in control experiments, how often do butterflies die? Are there offsite effects? It's striking that single-guide RNAs led to such strong effects. Is this common outside of this system? Is it possible to explore the function effects at the boundary element - are these generating large deletions (for example, like Mazo-Vargas et al., 2022)? For the mosaic experiments, how frequent are these effects in nature or captive stocks? Would it be possible to resequence these types of effects? At the moment, this data, while compelling, was hard to put into the context of the experiments above without understanding how common the effects are. Ideally, there would be resequencing of these tissues, which could be targeted, but it was not clear to me the general rates of these variants.

We agree with this assessment completely: mosaics complicate the proper interpretation of CRISPR based perturbation assays in regulatory regions. Here, unlike in Mazo-Vargas et al. (2022), we were unable to breed homeotic effects to a G1 generation, possibly because the phenotypes are dominant and lethal at the embryonic stage (see also our reply to Reviewer 2). This means that mosaic mutants are often survivors with clones of restricted size in the wing, and they are probably rare, but we are unable to meaningfully measure a mutation spectrum frequency (e.g. how often large deletions are generated). As mentioned in the first paragraph of our Discussion, we think that many of the phenotypes we observed (besides the Ubx GOF effects from the BE targeting) were confounded by alleles that could include large SVs. We aim to address these questions in an upcoming manuscript, at a locus where regulatory perturbation does not impact survival, including using germline mutants and unbiased genotyping (whole genome resequencing).

We elaborated on this issue in our Discussion:

"It is crucial here to highlight the limitations of the method, in order to derive proper insights about the functionality of the regulatory regions we tested. In essence, butterfly CRISPR experiments generate random mutations by non-homologous end joining repair, that are usually deletions (Connahs et al. 2019; Mazo-Vargas et al. 2022; Van Belleghem et al. 2023). Ideally, regulatory CRISPR-induced alleles require genotyping in a second (G1) generation to be properly matched to a phenotype (Mazo-Vargas et al. 2022). Possibly because of lethal effects, we failed to pass G0 mutations to a G1 generation for genotyping, and were thus limited here to mosaic analysis. As adult wings have lost scale building cells that may underlie a given phenotype, we circumvented this issue by genotyping a pupal forewing displaying an homeotic phenotype in the more efficient Antp-Ubx_BE perturbation experiment (Fig. S4). In this case, PCR amplification of a 600 bp fragment followed by Sanger sequencing recovered signatures of indel variants, with mixed chromatograms starting at the targeted sites. But in all other experiments (CRM11, IT1, and AS5’ targets), we did not genotype mutant tissues, as they were only detected in adult stages and generally with small clone sizes. Some of these clones may have been the results of large structural variants, as data from other organisms suggests that Cas9 nuclease targeting can generate larger than expected mutations that evade common genotyping techniques (Shin et al. 2017; Adikusuma et al. 2018; Kosicki et al. 2018; Cullot et al. 2019; Owens et al. 2019). Even under the assumption that such mutations are relatively rare in butterfly embryos, the fact we injected >100 embryos in each experiment makes their occurrence likely (Fig. 9), and we are unable to assign a specific genotype to the homeotic effects we obtained in CRM11, IT1 and AS5’ perturbation assays."

Our revision also includes a new Fig. S4 that features the mosaic genotyping of a G0 Antp-Ubx_BE mutant tissue. While this does not fully address the reviewer questions, it provides reasonable validation that the frequent GOF effects we observed upon perturbation at this target site are generated by on-target indels from DNA repair.

**Author response image 1. sa2fig1:** Validation of CRISPR-induced DNA Lesions in an Antp-Ubx_BE crispant pupat forewing. (A-A') Pupal forewing cuticle phenotype of an Antp-Ubx_BE J. coenia crispant, as in Fig. S3. (B-B") Aspect of the same forewing under trans-illumination following dissection out of the pupal case. Regions from mutant clones have a more transparent appearance. (C). Sanger sequencing of an amplicon targeting the Antp-Ubx_BE region in the mutant tissue shown in panel B", compared to a control wing tissue, showing mixed chromatogram around the expected CRISPR cutting site due to indel mutations from non-homologous end-joining.

In sum, I enjoyed the extensive mosaic perturbations. However, I feel that more molecular descriptions would elevate the work and make a larger impact on the field.
**Reviewer #2 (Public Review):**
Summary:The existence of hox gene complexes conserved in animals with bilateral symmetry and in which the genes are arranged along the chromosome in the same order as the structures they specify along the anteroposterior axis of organisms is one of the most spectacular discoveries of recent developmental biology. In brief, homeotic mutations lead to the transformation of a given body segment of the fly into a copy of the next adjacent segment. For the sake of understanding the main observation of this work, it is important to know that in loss-of-function (LOF) alleles, a given segment develops like a copy of the segment immediately anterior to it, and in gain-of-function mutations (GOF), the affected segment develops like a copy of the immediately posterior segment. Over the last 30 years the molecular lesions associated with GOF alleles led to a model where the sequential activation of the hox genes along the chromosome result from the sequential opening of chromosomal domains. Most of these GOF alleles turned out to be deletions of boundary elements (BE) that define the extent of the segment-specific regulatory domains. The fruit fly *Drosophila* is a highly specialized insect with a very rapid mode of segmentation. Furthermore, the hox clusters in this lineage have split. Given these specificities it is legitimate to question whether the regulatory landscape of the BX-C we know of in *D. melanogaster* is the result of very high specialization in this lineage, or whether it reflects a more ancestral organization. In this article, the authors address this question by analyzing the continuous hox cluster in butterflies. They focus on the intergenic region between the Antennapedia and the Ubx gene, where the split occurred in *D. melanogaster*. Hi-C and ATAC-seq data suggest the existence of a boundary element between 2 Topologically-Associated-Domain (TAD) which is also characterized by the presence of CTCF binding sites. Butterflies have 2 pairs of wings originating from T2 (forewing) specified by Antp and T3 specified by Ubx (hindwing). Remarkably, CRISPR mutational perturbation of this boundary leads to the hatching of butterflies with homeotic clones of cells with hindwings identities in the forewing (a posteriorly oriented homeotic transformation). In agreement with this phenotype, the authors observe ectopic expression of Ubx in these clones of cells. In other words, CRISPR mutagenesis of this BE region identified by molecular tool give rise to homeotic transformations directed towards more posterior segment as the boundary mutations that had been 1st identified on the basis of their posterior oriented homeotic transformation in *Drosophila*. None of the mutant clones they observed affect the hindwing, indicating that their scheme did not affect the nearby Ubx transcription unit. This is reassuring and important first evidence that some of the regulatory paradigms that have been proposed in fruit flies are also at work in the common ancestor to *Drosophilae* and Lepidoptera.Given the large size of the Ubx transcription unit and its associated regulatory regions it is not surprising that the authors have identified ncRNA that are conserved in 4 species of Nymphalinae butterflies, some of which also present in *D. melanogaster*. Attempts to target the promoters by CRISPR give rise to clones of cells in both forewings and hindwings, suggesting the generation of regulatory mutations associated with both LOF and GOF transformations. The presence of clones with dual homeosis suggests the targeting of Ubx activator and repression CRMs. Unfortunately, these experiments do not allow us to make further conclusions on the role of these ncRNA or in the identification of specific regulatory elements. To the opinion of this reviewer, some recent papers addressing the role that these ncRNA may play in boundary function should be taken with caution, and evidence that ncRNA(s) regulate boundaries in the BX-C in a WT context is still lacking.Strengths:The convincing GOF phenotype resulting from the targeting of the Antp-Ubx_BE.Weaknesses:The lack of comparisons with the equivalent phenotypes obtained in *D. melanogaster* with for example the Fub mutation.

We are grateful for this excellent contextualization of our findings and have incorporated some of the historical elements into our revision, as detailed below.

**Reviewer #2 (Recommendations For The Authors):**
In the whole paper, the authors bring the notion of boundaries through the angle of the existence of TADs and ignore almost entirely to explain the characteristics of boundary mutation in the BX-C. To my knowledge examples where targeted boundary deletions between TADs result in misregulation of the neighboring genes, and/or a phenotype, are extremely sparse (especially in the context of the mouse hox genes). Given the extensive litterature describing the boundary mutations and their associated GOF phenotypes, the paper would certainly gain strength if the authors justify their approach through this wealth of information. I must admit that this referee is surprised by the absence of any references to the founding work of the Karch and Bender laboratories on this topic. As a matter of fact, one of the founding members of the boundary class of regulatory elements was already brought in 1993 with the Fab-7 and Mcp elements of the BX-C. Based on gain-of-function homeotic phenotypes, additional Fab boundaries were added to the list. Finally, in 2013, Bender and Lucas (https://www.ncbi.nlm.nih.gov/pmc/articles/PMC3606092/) identified the Fub boundary element that delimits the Ubx and abd-A domains in the BX-C. Fub fulfills the criterium of lying at the border of 2 neighboring TADs. Significantly, a deletion of Fub leads to a very penetrant and strong homeotic gain-of-function phenotype in which the flies hatch with a 1st abdominal segment transformed into the 2nd. In agreement with this, abd-A is expressed one parasegment too anterior in embryos. This is exactly the observation gathered from the targeted mutations in the Antp-Ubx_BE; a dominant transformation of anterior to posterior wing accompanied by an ectopic expression of Ubx in the forming primordia of the forwing where it is normally silenced. I believe the paper would gain credibility if the results were reported with the knowledge of the similarities with Fub.Line 53, I am not aware of the existence of TADs for each of the 9 regulatory domains. The insulators delimit the extent of the regulatory domains but certainly not of TADs.

We thank the reviewer for these suggestions, as well as for the correction – we agree our previous text suggested that all BX-C boundaries are TAD boundaries, which was incorrect. We added a new introduction paragraph that combines classic literature on GOF mutations at boundary elements with recent evidence these are TAD insulators, including Fub (as suggested), and adding Fab-7 for breadth of scope.

"For instance, the deletion of a small region situated between Ubx and abd-A produces the Front-ultraabdominal phenotype (Fub) where the first abdominal segment (A1) is transformed into a copy of the second abdominal segment A2, due to a gain-of-expression of abd-A in A1 where it is normally repressed (Bender and Lucas 2013). At the molecular level, the Fub boundary is enforced by insulating factors that separate Topologically Associating Domains (TADs) of open-chromatin, while also allowing interactions of Ubx and abd-A enhancers with their target promoters (Postika et al. 2018; Srinivasan and Mishra 2020). Likewise, the Fab-7 deletion, which removes a TAD boundary insulating abd-A and Abd–B (Moniot-Perron et al. 2023), transforms parasegment 11 into parasegment 12 due to an anterior gain-of-expression of Abd-B (Gyurkovics et al. 1990). By extrapolation, one may expect that if the *Drosophila* Hox locus was not dislocated into two complexes, Antp and Ubx 3D contact domains would be separated by a Boundary Element (BE), and that deletions similar with Fub and Fab-7 mutations would result in gain-of-function mutations of Ubx that could effectively transform T2 regions into T3 identities."

A reference to the 1978 Nature article of Lewis should be added after line 42 of introduction.

Added

Line 56-57; the BX-C encoded miRNAs are known to regulate Ubx and abd-A, but not Abd-B.

Corrected

From lines 57 to 61, the authors mention reports aimed at demonstrating a role of ncRNA into Ubx regulation. To my eyes, these gathered evidences are rather weak. A reference to the work of Pease et al in Genetics in 2013 should be mentioned (https://www.ncbi.nlm.nih.gov/pmc/articles/PMC3832271/).

Added. Our paragraph includes qualifier language about the functionality of the Ubx-related ncRNAs (“are thought to”, “appears to”), and updated references regarding bxd (Petruk et al. 2006; Ibragimov et al. 2023).

Line 62 authors, should write "Little is known about how Hox genes are regulated outside of *Drosophila*" and not flies.

Corrected

Lines 110-112 could lncRNA:Ubx-IT1 correspond to PS4 antisense reported by Pease et al in 2023 (see URL above)? Lines 115-117, could lncRNA:UbxAS5' correspond to bxd antisense of Pease et al in 2023 (see above)?

As we could not detect sequence similarities, we preferred to avoid drawing homology, and we intentionally avoided reference to the fly transcripts when we named IT1 and AS5’. This said, we agree it is important to clarify that further studies are needed to clarify this relationship. We elaborated on this point in our discussion:

"Of note, a systematic in-situ survey (Pease et al. 2013) showed that *Drosophila* embryos express an antisense transcripts in its 5’ region (lncRNA:bxd), as well as within its first intron (lncRNA:PS4). It is thought that *Drosophila* bxd regulates Ubx, possibly by transcriptional interference or by facilitation of the Fub-1 boundary effect (Petruk et al. 2006; Ibragimov et al. 2023), while the possible regulatory roles of PS4 remain debated (Hermann et al. 2022). While these dipteran non-coding transcripts lack detectable sequence similarity with the lepidopteran IT1 and AS5’ transcripts, further comparative genomics analyses of the Ubx region across the holometabolan insect phylogeny should clarify the extent to which Hox cluster lncRNAs have been conserved or independently evolved."

Lines 154-155: "This concordance between Hi-C profiling and CTCF motif prediction thus indicates that Antp-Ubx_BE region functions as an insulator between regulatory domains of Antp and Ubx ». This is only correlative, I would write "suggests" instead of "indicates" and add a "might function".

Corrected as suggested.

Line 254, I assume the authors wish to write Ubx-IT1 in V. cardui instead of Ubx-T1.

Typo corrected

Line 255 : Fig.5 is absent from the pdf file and replaced by table 1. I did not find a legend for Table 1.

Corrected, with our sincere apologies for the loss of this image in our first submission.

Line 293 "Individual with hindwing clones 2.75 times more common than...." "are" is missing?

Corrected

Lines 303-313, it is not entirely clear how many guide RNAs were injected. Would be useful to indicate the sites targeted in Fig.S8.

We specify in the revised text : using a single guide RNA (Ubx11b9)

Lines 323-337: it is not entirely clear to this referee (a drosophilist) if those spontaneous mutations can be inbred or whether these individuals are occasional mosaics. In general, did anyone try to derive lines from those mosaic animals? Is it possible to hit the germline at the syncitial stages at which the guides are injected? Are the individuals with wing phenotype fertile? Given the fact that the Antp-Ubx_BE mutations should be dominant, I wonder if this characteristic would not help in identifying germline transmission. Similar remark for the discussion where the authors explain at line 360, that genotyping can only be done in the progeny of the Go. I do not have the impression that the authors have performed this genotyping and if I am right, I do not understand why.

We improved our discussion section on this topic (new text in orange):

"It is crucial here to highlight the limitations of the method, in order to derive proper insights about the functionality of the regulatory regions we tested. In essence, butterfly CRISPR experiments generate random mutations by non-homologous end joining repair, that are usually deletions (Connahs et al. 2019; Mazo-Vargas et al. 2022; Van Belleghem et al. 2023). Ideally, regulatory CRISPR-induced alleles require genotyping in a second (G1) generation to be properly matched to a phenotype (Mazo-Vargas et al. 2022). Possibly because of lethal effects, we failed to pass G0 mutations to a G1 generation for genotyping, and were thus limited here to mosaic analysis. As adult wings have lost scale building cells that may underlie a given phenotype, we circumvented this issue by genotyping a pupal forewing displaying an homeotic phenotype in the more efficient Antp-Ubx_BE perturbation experiment (Fig. S4). In this case, PCR amplification of a 600 bp fragment followed by Sanger sequencing recovered signatures of indel variants, with mixed chromatograms starting at the targeted sites. But in all other experiments (CRM11, IT1, and AS5’ targets), we did not genotype mutant tissues, as they were only detected in adult stages and generally with small clone sizes. Some of these clones may have been the results of large structural variants, as data from other organisms suggests that Cas9 nuclease targeting can generate larger than expected mutations that evade common genotyping techniques (Shin et al. 2017; Adikusuma et al. 2018; Kosicki et al. 2018; Cullot et al. 2019; Owens et al. 2019). Even under the assumption that such mutations are relatively rare in butterfly embryos, the fact we injected >100 embryos in each experiment makes their occurrence likely (Fig. 9), and we are unable to assign a specific genotype to the homeotic effects we obtained in CRM11, IT1 and AS5’ perturbation assays."

We agree that the work we conducted with mosaics has important caveats. So far, our attempts at breeding homeotic G0 mutants have not been fruitful at this locus, while less deleterious loci can yield viable alleles into further generations, such as WntA (published) and cortex (in prep.). We prefer to stay vague about negative data here, as it is difficult to disentangle if they were due to real mutational effects (e.g. the alleles can be dominant and lethal in the G1 generation) to failure to germline carriers of mutations as founders, or to health issues that are often amplified by inbreeding depression (including a possible iflavirus in our V. cardui cultures).

We concur with the prediction that Antp-Ubx_BE mutations are probably dominant, and intend to follow up with similar GOF experiments in the Plodia pantry moth, a laboratory model for lepidopteran functional genomics that is more amenable than butterflies to inbreeding and long-term studies in mutant lines. In our experience (https://www.frontiersin.org/articles/10.3389/fevo.2021.643661/full), Ubx coding knock-out can be more extensive in Plodia than in butterflies, so we think these animals will also be more resilient to the deleterious effects of the GOF phenotype.

Line 423, 425, I am not a fan of the term "de-insulating!!!!!

We replaced this neologism by Similar deletion alleles resulting in a TAD fusion and misexpression effect (see below).

Line 425, why bring the work on Notch while there are so many examples in the BX-C itself....

Our revised sentence makes it more clear we are referring here to documented examples of deletion-mediated TAD fusion (ie. featuring a conformation capture assay such as HiC/micro-C):

This suggests a possible loss of the TAD boundary in the crispant clones, resulting in a TAD fusion or in a long-range interaction between a T2-specific enhancer and Ubx promoter. Similar deletion alleles resulting in a TAD fusion and misexpression effect have been described at the Notch locus in *Drosophila* (Arzate-Mejía et al. 2020), in digit-patterning mutants in mice and humans (Lupiáñez et al. 2015; Anania et al. 2022), or at murine and fly Hox loci depleted of CTCF-mediated regulatory blocking (Narendra et al. 2015; Gambetta and Furlong 2018; Kyrchanova et al. 2020).

Our revision also includes more emphasis on the *Drosophila* BX-C boundary elements Fub and Fab-7 (see above).